# Self-Guided Hierarchical Exploration for Generalist Foundation Model Web Agents

**Qianlan Yang**
University of Illinois Urbana Champaign
qianlan2@illinois.edu

**Xiangjun Wang**
Amazon
xjai@amazon.com

**Danielle Perszyk**
Amazon
perszdan@amazon.com

**Yu-Xiong Wang**
University of Illinois Urbana Champaign
yxw@illinois.edu

## Abstract

Foundation models have recently shown strong potential as web agents, capable of interpreting high-level instructions and interacting with complex web interfaces. However, existing training paradigms for these agents often rely on predefined task datasets and curated demonstrations, limiting their scalability, adaptability, and capacity for self-improvement. In this work, we introduce *Self-guided hierArchical exploration for Generalist wEb agents* (SAGE), a new training framework designed to support autonomous skill acquisition through self-guided hierarchical exploration. Our method introduces a three-tier exploration strategy: a pre-exploration phase to build structural understanding of web environments, a top-level exploration strategy to generate a self-evolving curriculum of tasks from easy to hard, and a low-level exploration mechanism that combines planning-based rollouts with step-wise learning to improve policy efficiency. Together, these components form a scalable, supervision-free framework for web agent training. Experimental results on WebVoyager and WebArena demonstrate that our method significantly outperforms prior approaches, enabling foundation model agents to learn complex web tasks with greater generalization and robustness. Our project can be found at https://yanqval.github.io/SAGE/.

## 1 Introduction

Foundation models have demonstrated impressive generalization across domains such as natural language processing, code generation, and multimodal reasoning [1–3]. With their strong capabilities in understanding and generating structured content, foundation models, particularly vision-language models (VLMs) have recently shown promise as web agents, capable of interpreting high-level instructions and interacting with visual user interfaces [4, 5]. These agents have the potential to automate real-world tasks on the web, such as booking tickets, extracting information, and navigating software systems, without domain-specific engineering or fine-tuning for each application.

Despite this promise, existing approaches to training foundation model web agents face significant limitations. Most methods rely on static datasets composed of human-authored tasks and curated expert demonstrations [6–9], which are expensive to collect and fail to capture the long-tail complexity of real web interactions. Furthermore, these methods often lack mechanisms for continuous self-improvement, making it difficult for agents to scale their capabilities beyond the data they were trained on.

39th Conference on Neural Information Processing Systems (NeurIPS 2025).

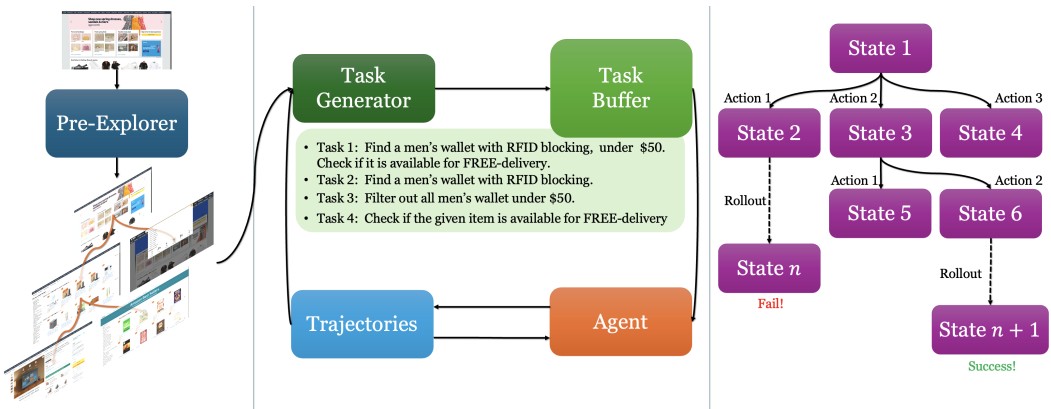

Figure 1: **Framework of SAGE. Left**: In the pre-exploration phase, the agent automatically collects structural information from the environment. **Middle**: In the top-level exploration phase, the task generator proposes new tasks based on the agent's performance in the previous iteration. **Right**: In the low-level exploration phase, the agent uses MCTS to explore and solve individual tasks efficiently. We also provided a case study to show SAGE works in Appendix Section F.

To address these challenges, we propose a new training framework for generalist foundation model web agents that emphasizes *self-guided exploration* as the core learning principle. Rather than depending on fixed task templates or expert demonstrations, our framework enables agents to autonomously explore real-world websites, generate their own instructional goals, and acquire skills through structured interaction. At the heart of our framework is a *three-stage exploration hierarchy* that supports curriculum-driven learning from the ground up.

As illustrated in Figure. 1, we begin with a *pre-exploration phase*, in which the agent autonomously traverses the structure of the website before training begins. This phase allows the agent to construct a semantic map of the environment, by capture page layouts, interactive elements, and navigation paths, which serves as the foundation for later skill generation. Next, we introduce a *top-level exploration strategy* that governs how instructional tasks are proposed and evolved throughout training. By analyzing past performance, this component adaptively generates new tasks from easy to hard by composing or decomposing instruction templates, thus enabling a adaptive learning curriculum. Finally, we develop a *low-level exploration strategy* to enhance the efficiency of within-task learning. Using Monte Carlo Tree Search (MCTS) [10] and a multi-action agent policy, the agent explores promising action sequences for each task and refines its behavior through structured rollout and preference-based learning.

Together, these components form an end-to-end system *Self-guided hierArchical exploration for Generalist wEb agents* (SAGE), in which agents progressively discover how to interact with and reason about complex web environments. Our experiments demonstrate that this adaptive exploration framework leads to stronger generalization and performance compared to prior approaches that rely on static task definitions or imitation data. By enabling agents to learn from their own interaction cycles, our method takes a step toward scalable, self-improving web agents grounded in foundation model capabilities.

Our **Contributions** are three-fold. **(i)** We propose a top-level exploration strategy that enables agents to autonomously generate and adapt instructional tasks from easy to hard, forming a self-evolving curriculum without relying on predefined task sets. **(ii)** We introduce a low-level exploration strategy that combines MCTS-based planning and stepwise learning, allowing agents to efficiently discover viable action sequences and improve under sparse rewards. **(iii)** We unify all components into SAGE, a generalist web agent training framework that leverages multi-level exploration to progressively acquire skills and generalize across complex, real-world internet environments. As demonstrated by our empirical results, SAGE achieves state-of-the-art performance on web-based tasks. On WebArena [11], it outperforms all open-source baselines by 26%, and even surpasses proprietary models by 11%.

## 2   Related Work

**Foundation Model Web/GUI Agents.** The emergence of Large Language Models (LLMs) and Vision-Language Models (VLMs) has spurred the development of agents capable of interacting with structured interfaces such as web pages, software GUIs, and mobile applications [1, 4, 12, 13]. Prior approaches generally fall into two groups: training-free systems that rely on prompt engineering and orchestration of frozen models [3, 5, 14], and fine-tuning-based methods that adapt model parameters using supervised trajectories [6–9]. While training-free systems offer flexibility, they are often bounded by the static capabilities of the underlying models. On the other hand, supervised fine-tuning improves task-specific performance but typically depends on curated task datasets and lacks adaptability. To mitigate this, recent methods introduce automated evaluators that estimate task completion as a reward signal [15, 16], or propose instruction-based self-supervision [17]. However, most of these pipelines still rely on static or benchmark-defined instruction pools. In contrast, our method introduces a structured, hierarchical exploration framework that adaptively generates and organizes tasks from easy to hard, enabling agents to acquire skills and improve autonomously without relying on static benchmarks or curated supervision.

**Reinforcement Learning for LLMs.** Reinforcement learning (RL) has increasingly been applied to align LLM behaviors with downstream tasks, especially in domains that require sequential decisions and delayed feedback. While RL was initially used to fine-tune models based on human preferences or scalar rewards, recent work has shown that it can be extended to environments with interface-level interactions [13, 18]. Actor-critic methods and policy optimization frameworks such as Proximal Policy Optimization [19] and Direct Preference Optimization (DPO) [20] have proven effective for training agents to operate over complex action spaces. However, most existing methods assume a fixed task distribution and lack a mechanism for adaptively shaping the learning trajectory. Our work complements this line by introducing a self-adaptive task curriculum that evolves in response to the agent's learning progress, enabling continual policy improvement under a changing task landscape.

**Instruction Generation and Skill Discovery.** Generating training signals autonomously has emerged as a key mechanism for improving the scalability and generality of foundation model agents. Instruction generation approaches such as Self-Instruct [21] and Meta-Prompting [22] enable models to bootstrap themselves by producing and solving their own tasks, while others fine-tune models in simulated planning environments [23]. Parallel work in unsupervised reinforcement learning has focused on discovering diverse skills without access to task labels [24, 25], but these approaches often prioritize behavioral diversity over goal-directed utility. In the domain of web agents, recent work PAE [26] attempts to train agents via instruction sets either statically defined or updated from interaction history, but still relies on initial human-written instructions to define the scope of training. Our method departs from this design by constructing a fully self-evolving task curriculum that requires no external instruction corpus and enables agents to grow their capabilities from primitive operations to compound, high-level behaviors through direct interaction.

## 3   Background

**Problem fomulation.**   We formalize the training of generalist web agents as a contextual Markov Decision Process (MDP) defined by the tuple $\mathcal{M} = (\mathcal{S}, \mathcal{A}, \mathcal{T}, \mathcal{R}, H, \mathcal{I})$, where $\mathcal{S}$ is the state space, $\mathcal{A}$ the action space, $\mathcal{T}$ the environment dynamics, $\mathcal{R}$ the reward function, $H$ the episode horizon, and $\mathcal{I}$ the task instruction space.

At the start of each episode, a task instruction $I \in \mathcal{I}$ is sampled—typically a natural language command (e.g., "add the product to the cart"). The agent begins interacting with a web environment and receives a state $s_t \in \mathcal{S}$ at each timestep $t \in \{1, \ldots, H\}$, comprising the current webpage content and interaction history. Conditioned on $s_t$ and $I$, the agent selects an action $a_t \in \mathcal{A}$, such as clicking or typing. The environment transitions to $s_{t+1} \sim \mathcal{T}(s_{t+1} \mid s_t, a_t)$, and the process continues.

The agent follows a policy $\pi_\theta(a_t \mid s_t, I)$, parameterized by $\theta$, and receives a scalar reward at the end of the episode indicating task success. Crucially, the true instruction distribution $\mathcal{I}$ is unknown during training, and the agent must learn from proxy tasks or self-generated curricula. The objective is to maximize expected return over task and trajectory distributions:

$$\max_{\theta} \ \mathbb{E}_{I \sim \mathcal{I}, \, (s_1, a_1, \ldots, s_H) \sim \pi_\theta} \left[ \sum_{t=1}^{H} r(s_t, a_t, I) \right].$$

**Multimodal Environments.** To enable lightweight and efficient training for interactive web agents, we build upon the web environment developed in PAE [17], which integrates both visual and structural information. In the original PAE setup, agents receive a screenshot of the current webpage annotated with *set-of-marks* that help the agent precisely identify interactive elements. Each marked element is paired with a textual description to provide semantic grounding for action decisions.

We further extend this environment by incorporating the webpage's accessibility tree as an additional input modality. The accessibility tree captures the full structure of all interactive elements in a hierarchical form, including those not explicitly annotated in the screenshot. To bridge the visual and structural modalities, we establish a one-to-one alignment between the set-of-marks labels and corresponding nodes in the accessibility tree. This alignment allows the agent to use consistent numerical identifiers to reference web elements across both visual and semantic representations, enabling seamless cross-modal reasoning and interaction. We provided a more detailed setting of our environment in Appendix Section B.

## 4 Method

In this section, we propose SAGE, a new training framework for developing generalist foundation model-based web agents that can adaptively explore and interact with real-world Internet environments, while continuously self-improving in a self-guided manner. Our objective is to enable agents not only to complete individual tasks but also to continually acquire new skills through experience—starting from simple goals and gradually advancing to more complex, real-world objectives. At the core of our method is a hierachical exploration framework where the agent is self-guided in understanding its environment, generating meaningful tasks for practice, and discovering effective strategies for solving them.

To support this process, we introduce three levels of exploration, each operating at a different functional scope: (1) a *pre-exploration* strategy, which enables the agent to build an initial understanding of the workspace and identify the interactive structure of the environment; (2) a *top-level exploration* strategy, which helps the agent discover and organize tasks from easy to hard, establishing a structured path for learning and skill acquisition; and (3) a *low-level exploration* strategy, which allows the agent to efficiently explore fine-grained actions within a specific task, thereby enhancing its ability to solve concrete objectives. Finally, we conclude this section with an overview of how these components integrate into a unified framework for scalable skill learning and generalization. The entire framework of SAGE is illustrated in Figure 1. We also provided additional algorithm details in Appendix Section A.

### 4.1 Pre-Exploration Strategy

To enable the agent to develop an initial understanding of a website's structure and functionality prior to training, we introduce a *pre-exploration* phase. The goal of this phase is to allow the agent to autonomously investigate the interactive landscape of the environment—identifying key UI elements, navigation paths, and potential functionality—so that it may later generate and reason about feasible tasks relevant to the site.

In this stage, the agent receives only the URL of the website's homepage as input. Without any task-specific instruction or supervision, the agent begins exploring the site from this entry point using a Breadth-First Search (BFS) strategy. We maintain a queue of *states*, where each state corresponds to a unique URL discovered during the exploration. Initially, the queue contains only the homepage URL. At each step, the agent dequeues a state and decides whether the page warrants further action, especially if it has been previously visited. If so, the agent selects up to $k$ actions that are likely to lead to unexplored or semantically distinct pages. These actions are executed, and the resulting new URLs are enqueued as candidate states for further exploration. The process continues until the queue is empty or a predefined limit on the total number of states is reached.

Throughout this exploration, the agent logs detailed information about the discovered environment. Specifically, it records: (1) the screenshot of each visited state; (2) structural transitions between states, described in a textual format such as "page A transitions to page B via action `click [Search]`"; and (3) page-specific textual metadata, such as product categories on commercial sites or section labels on documentation pages. These collected data form a knowledge base of the environment, which will be used by subsequent modules to generate tasks and guide agent learning.

## 4.2 Top-Level Exploration Strategy

To guide the agent toward learning complex tasks in a structured and feasible manner, we introduce a *top-level exploration strategy* that selects and adapts tasks dynamically across training iterations. This module is composed of a self-adaptive task generator and a task replay buffer, working together to provide an evolving curriculum that matches the agent's current capabilities.

During the initial training iteration, the task generator relies exclusively on the knowledge acquired in the pre-exploration phase. It samples tasks grounded in the observed website structure and content, such as retrieving product information, locating links, or filling out forms. In subsequent iterations, the generator is conditioned not only on the pre-exploration information but also on the agent's past execution trajectories, particularly their success and failure patterns. Based on this history, the generator proposes new tasks using three complementary strategies.

First, similar to the initial iteration, the generator continues to introduce new tasks derived from the structural information collected during pre-exploration, expanding the task space beyond previously attempted goals. Second, the generator identifies tasks with low success rates, typically those that the agent struggles to complete and produces simplified variants to help the agent learn more incrementally. For example, a complex instruction like "What is the price of the table which I bought yesterday?" may be decomposed into subtasks such as "What is the table I bought yesterday?" and "What is the price of this given item?". In other cases, the generator may relax constraints from a difficult task to form a more tractable one. For instance, simplifying "Find a recipe of apple pie which needs less than 30 minutes, has more than 4 stars, and at least 50 likes" to "Find a recipe of apple pie which needs less than 30 minutes and at least 50 likes." Third, for tasks that the agent consistently solves correctly, the generator increases difficulty by proposing more challenging variants. These may be constructed by composing simpler instructions into a compound task, or by tightening the conditions in the original query, thereby advancing the curriculum.

**Task Buffer.** To store and manage the generated tasks across iterations, we employ a double-buffer replay system. The first buffer contains only tasks proposed during the initial iteration, preserving a stable sample of the original task distribution. The second buffer stores tasks generated in later iterations as part of the self-adaptive curriculum. At training time, tasks are sampled from both buffers: a portion is drawn from the initial buffer to retain exposure to foundational skills, while the remainder is drawn from the adaptive buffer to promote progressive learning. As new tasks are generated, they randomly replace entries in the second buffer, ensuring freshness while bounding the total number of tasks considered in any given iteration.

This replay mechanism balances the agent's exposure to both early-stage and evolving tasks, ensuring that its foundational abilities are retained while new capabilities are continually acquired. By dynamically adjusting task difficulty based on performance feedback, this top-level exploration strategy enables the agent to follow a self-paced learning trajectory, from mastering basic interactions to solving complex, real-world tasks.

## 4.3 Low-Level Exploration Strategy

To enable efficient decision-making during the execution of individual tasks, we introduce a *low-level exploration strategy* based on Monte Carlo Tree Search (MCTS) [10]. This component improves the agent's ability to explore diverse action sequences, reason over possible outcomes, and prioritize high-potential decisions, which is particularly important in sparse-reward environments with long task horizons.

Instead of rolling out trajectories linearly from the initial state to the end of an episode, the agent performs structured exploration using MCTS. Each search begins with a root node representing the task's starting state, which is typically defined by the entry URL. At each iteration, the agent traverses

the tree by selecting child nodes estimated to maximize the likelihood of task success, continuing until a leaf node is reached. This leaf is then expanded using up to $k$ candidate actions proposed by the agent, resulting in $k$ new states. Based on the estimated success probability, the most promising of thes is selected to grow the tree. From the final candidate node, the agent conducts a standard rollout using the remaining step budget of the episode. The outcome of this rollout (success or failure) is recorded and backpropagated through the tree to refine future selection decisions. This iterative process enables the agent to concentrate exploration on high-potential trajectories, improving task success rates while reducing inefficient exploration.

**Multi-Action Agent Policy.** To support both tree-based planning and linear rollouts, we design a *multi-action agent policy*. Given a single state, the policy outputs up to $k$ candidate actions, each associated with a confidence score. These candidates are used differently depending on context. During MCTS-based rollout, all candidate actions are treated equally, regardless of their confidence, to encourage broader exploration. During standard execution (i.e., non-tree rollout), the agent selects a single action according to the highest-confidence prediction. This dual-mode usage enables the policy to be both exploratory and decisive, depending on the operational stage.

**Training-Based Outcome Evaluator.** We adopt a sparse reward structure in which the agent receives a binary reward only at the end of a trajectory, indicating whether the task was successfully completed. To evaluate this outcome reliably, we introduce a learned outcome evaluator. Similar to prior approaches [17, 15], the evaluator takes as input the task instruction, the agent's final output, and the history of the last three steps leading to the final state. This condensed representation captures the agent's recent reasoning and actions. The evaluator is trained using both model-generated trajectories and a limited number of human-annotated examples, allowing it to generalize effectively across a wide range of task formats. A well-trained evaluator improves the accuracy of success detection and serves as a reliable signal for downstream learning.

**Stepwise Learning Strategy.** To update the agent's policy based on observed success and failure cases, we adopt a learning approach inspired by the stepwise variant of Direct Preference Optimization (Step-DPO) [20, 27]. Given a pair of trajectories, one successful and one failed, we identify the divergence point at state $s_k$, where the two trajectories share a common prefix up to $s_{k-1}$. The policy is then trained to prefer the successful continuation from that point onward. Our use of MCTS facilitates the collection of diverse trajectory pairs by sampling multiple decision paths during tree expansion and rollout. This provides ample training signal for the Step-DPO objective and enables the agent to improve policy performance through fine-grained, localized credit assignment.

Together, the low-level MCTS planner, the multi-action policy, the outcome evaluator, and the preference-based learning strategy form a coherent execution framework. These components enable the agent to efficiently explore within-task decisions, accurately evaluate outcomes, and learn from partial successes and failures, thereby accelerating the acquisition of robust web-interaction skills.

## 4.4 Overall Framework

Inspired by prior work such as PAE [17], we adopt a modular training pipeline that integrates the components described above into a unified system. While PAE establishes a general iterative self-training structure, it relies on a static instruction distribution without mechanisms for adapting task difficulty or exploration strategy. Building upon this paradigm, our framework introduces *two key modifications*. First, we explicitly incorporate a pre-exploration phase before training begins, allowing the agent to construct a structured understanding of the website environment. Second, we build a self-guided learning loop that interleaves curriculum-driven task generation, efficient planning-based execution, and preference-based policy updates, enabling continuous adaptation and improvement.

Each training iteration proceeds as follows. We begin by using the top-level exploration strategy to propose new tasks based on the agent's current performance and the structural knowledge obtained from the pre-exploration phase. These tasks are added to the replay buffer, which maintains a mix of early-stage and adaptively generated goals. Next, a batch of tasks is sampled from the replay buffer, and the agent executes them using the low-level exploration strategy. This execution involves MCTS-based planning guided by the multi-action policy, as well as evaluation using the learned outcome model. Finally, the collected trajectories from these rollouts are used to update the policy via a stepwise learning algorithm inspired by Step-DPO. This cycle is repeated across iterations,

| | | Allrecipes | Amazon | Apple | ArXiv | GitHub | ESPN | Coursera |
|---|---|---|---|---|---|---|---|---|
| *Proprietary* | Claude 3.5 Sonnet | 50.0 | 68.3 | 60.4 | 46.5 | 58.5 | 27.3 | 78.6 |
| *Open-source* | Qwen2.5VL-7B | 0.0 | 2.3 | 4.5 | 2.3 | 0.0 | 0.0 | 2.3 |
| | Qwen2.5VL-32B | 2.3 | 18.4 | 17.5 | 6.2 | 4.7 | 0.0 | 7.4 |
| | LLaVa-7B | 0 | 0 | 0 | 0 | 0 | 0 | 0.0 |
| | LLaVa-34B | 0 | 0 | 2.3 | 0 | 2.4 | 0 | 0.0 |
| *Fine-Tuning* | LLaVa-7B PAE | 14.3 | 37.5 | 17.5 | 19.0 | 14.6 | 0.0 | 33.3 |
| | LLaVa-7B SAGE | 16.6 | 32.9 | 22.5 | 21.2 | 15.3 | 6.3 | 36.2 |
| | LLaVa-34B PAE | 22.7 | 53.7 | 38.5 | 25.6 | 14.6 | 13.6 | 42.9 |
| | LLaVa-34B SAGE | 28.4 | 57.9 | 42.7 | 29.3 | 18.2 | **18.7** | 46.1 |
| | Qwen2.5VL-7B PAE | 28.7 | 37.2 | 19.9 | 25.3 | 17.9 | 9.0 | 37.5 |
| | Qwen2.5VL-7B SAGE | 34.5 | 40.8 | 24.7 | 28.9 | 20.4 | 12.2 | 40.3 |
| | Qwen2.5VL-32B PAE | 40.2 | 59.5 | 45.1 | 30.3 | 39.8 | 16.4 | 49.7 |
| | Qwen2.5VL-32B SAGE | **43.6** | **64.7** | **51.8** | **35.0** | **56.2** | 18.2 | **62.4** |

| | | Cambridge Dictionary | BBC News | Google Maps | Google Search | HuggingFace | Wolfram Alpha | Average |
|---|---|---|---|---|---|---|---|---|
| *Proprietary* | Claude 3.5 Sonnet | 86.0 | 36.6 | 58.5 | 30.2 | 44.2 | 66.7 | 50.5 |
| *Open-source* | Qwen2.5VL-7B | 7.3 | 0.0 | 0.0 | 2.4 | 0.0 | 4.8 | 1.9 |
| | Qwen2.5VL-32B | 48.3 | 4.6 | 14.2 | 8.3 | 4.6 | 12.6 | 11.3 |
| | LLaVa-7B | 0 | 0 | 0 | 0 | 0 | 0 | 0.0 |
| | LLaVa-34B | 0 | 2.3 | 0 | 2.3 | 2.3 | 0 | 0.9 |
| *Fine-Tuning* | LLaVa-7B PAE | 52.4 | 18.6 | 22.5 | 23.3 | 19.0 | 24.4 | 22.3 |
| | LLaVa-7B SAGE | 68.0 | 24.3 | 24.3 | 25.0 | 21.5 | 24.4 | 26.0 |
| | LLaVa-34B PAE | 74.4 | 39.0 | 22.0 | 18.6 | 25.6 | 42.9 | 33.0 |
| | LLaVa-34B SAGE | 80.5 | 41.7 | 28.9 | 30.7 | 28.4 | 48.6 | 37.6 |
| | Qwen2.5VL-7B PAE | 71.3 | 16.3 | 12.9 | 4.5 | 16.3 | 38.3 | 25.0 |
| | Qwen2.5VL-7B SAGE | 75.2 | 22.8 | 20.6 | 12.7 | 22.1 | 45.4 | 30.1 |
| | Qwen2.5VL-32B PAE | 77.4 | 37.5 | 30.1 | 26.4 | 30.9 | 55.8 | 40.8 |
| | Qwen2.5VL-32B SAGE | **83.6** | **42.8** | **34.1** | **37.1** | **34.8** | **64.5** | **47.5** |

Table 1: **Results on WebVoyager.** Success rates reported in the table reflect full task completion rather than individual step accuracy. Each row corresponds to a specific website, and the *Average* column indicates the overall success rate across all tasks. The results demonstrate that SAGE consistently improves agent performance across diverse open-internet web environments.

enabling the agent to continuously refine its skills, explore new tasks, and improve performance in a self-guided manner.

# 5 Experiments

We design a series of experiments to evaluate the effectiveness of our adaptive exploration framework for training generalist foundation model web agents. Specifically, we aim to answer the following key questions: **(1)** Can the agent acquire meaningful web interaction skills from scratch, without any human-annotated demonstrations or task supervision? **(2)** How effectively can the agent solve complex, multi-step tasks that require conditional reasoning? **(3)** What is the contribution of each component in our framework to the overall performance? **(4)** How does the hierarchical exploration strategy—comprising pre-exploration, top-level task generation, and low-level rollout—impact the agent's ability to learn and generalize web tasks? We leave some additional experiments in Appendix Section C.

## 5.1 Environments

We evaluate on two realistic web-based benchmarks: **WebVoyager** [5] and **WebArena** [11].

**WebVoyager** consists of 643 instructional tasks spanning 15 real-world websites, including platforms such as Amazon and GitHub. Following the setup in PAE [17], we exclude two domains Google Flights and Google Bookings due to major structural changes that render the original tasks unachievable. As a result, we evaluate on the remaining 13 websites, comprising a total of 557 tasks.

**WebArena** is a closed-domain benchmark hosted on internal servers, featuring five realistic web environments: OpenStreetMap (Map), Reddit, GitLab, a content management system (CMS) for an online store, and OneStopShop (OSS). While WebArena originally included 812 tasks, many are unsuitable for training autonomous web agents due to execution constraints. For a fair comparison, we follow existing practice and adopt a combined task set drawn from WebArena-Easy [17] and WebArena-Lite [28] as our evaluation suite.

## 5.2 Baseline Comparisons

In this paper, we compare our method against two major categories of baseline models: (1) proprietary VLMs and (2) state-of-the-art open-source VLMs. For the first category, we use **Claude 3.5 Sonnet** [29] as a representative proprietary model. It is prompted using the same setup as prior work [5], including set-of-marks augmented screenshots and chain-of-thought action outputs.

For the open-source category, we evaluate several strong VLMs, including **LLaVa-7B/34B**, **Qwen2.5VL-7B/32B** [30, 31]. Within this category, we include three training variants: (1) **SFT-based** models, which are fine-tuned on trajectories generated by Claude 3 Sonnet over tasks proposed in PAE [17]; (2) models trained using the **PAE** framework [17]; and (3) models trained with **our proposed method**, which introduces multi-level adaptive exploration and self-supervised learning without relying on pre-defined instruction sets. This comparison allows us to assess the contributions of our exploration-driven framework relative to imitation-based and instruction-guided baselines across both closed- and open-domain web interaction tasks.

## 5.3 Main Results

As shown in the main experimental results presented in Tables 1 and 2, our method consistently outperforms strong baselines across both WebVoyager and WebArena. On WebVoyager, SAGE achieves a notable performance improvement of 8%–17% over PAE when using the same underlying vision-language model, demonstrating the benefit of our adaptive exploration framework in open-domain, real-world websites. In WebArena—featuring server-hosted, closed-domain environments with more structured and rigid interfaces—SAGE continues to deliver strong gains, outperforming PAE by 19%–256% across multiple model backbones.

In addition to outperforming PAE, we observe that SAGE consistently improves the performance of base models relative to their original, untrained versions. Notably, Qwen2VL-72B trained under our framework matches or even exceeds the performance of the proprietary Claude 3.5 Sonnet model in several benchmark scenarios. This finding underscores the potential of large open-source VLMs when coupled with structured, self-supervised training pipelines such as ours—highlighting the scalability and accessibility of our approach across different model sizes and training setups.

|  |  | Map | Reddit | OSS | Gitlab | CMS | Avg |
|---|---|---|---|---|---|---|---|
| *Proprietary* | Claude 3.5 Sonnet | 40.1 | 46.9 | 42.0 | 24.2 | 25.1 | 35.6 |
| *Open-source* | LLaVa-7B | 0.0 | 0.0 | 0.0 | 0.0 | 0.0 | 0.0 |
|  | LLaVa-34B | 1.3 | 0.0 | 0.0 | 0.0 | 0.0 | 0.6 |
|  | Qwen 2.5VL-7B | 0.9 | 10.2 | 20.2 | 8.3 | 7.4 | 9.4 |
|  | Qwen 2.5VL-32B | 13.4 | 18.4 | 26.4 | 11.6 | 8.3 | 15.6 |
| *Fine-Tuning* | Qwen 2.5VL-7B PAE | 23.8 | 24.6 | 37.4 | 15.3 | 12.4 | 22.7 |
|  | Qwen 2.5VL-7B SAGE | 28.1 | 29.7 | 43.2 | 18.9 | 15.6 | 27.1 |
|  | Qwen 2.5VL-32B PAE | 34.5 | 35.9 | 41.0 | 24.1 | 22.0 | 31.5 |
|  | Qwen 2.5VL-32B SAGE | **42.3** | **43.7** | **46.1** | **31.4** | **35.2** | **39.7** |

Table 2: **Results on WebArena.** Success rates indicate full task completion across five sandboxed, server-hosted environments. Each column represents a distinct website domain, and the *Avg* column reports the weighted average success rate based on the number of tasks per domain. The results show that SAGE significantly improves performance across all model variants.

To further analyze model behavior across varying levels of task complexity, we categorize the tasks in WebArena into three difficulty levels based on the minimum number of steps required for completion, as determined by human annotators. Specifically, tasks requiring fewer than 7 steps are labeled *Easy* (25% of all tasks), those requiring between 7 and 15 steps are labeled *Medium* (54%), and those with more than 15 steps are categorized as *Hard* (21%). We evaluate model performance separately within each bucket to assess their ability to generalize to long-horizon tasks.

As shown in Table 3, performance generally declines as task complexity increases across all models. However, the relative improvement of SAGE over baselines becomes more pronounced on harder tasks. For instance, Qwen2.5VL-32B trained with our method achieves comparable performance to Claude 3.5 Sonnet on *Hard* tasks and significantly outperforms other baselines. These findings highlight the effectiveness of our hierarchical exploration framework, particularly the curriculum-aware task proposer and MCTS-guided

|  |  | Easy | Medium | Hard | Avg |
|---|---|---|---|---|---|
| *Proprietary* | Claude 3.5 Sonnet | 65.0 | 30.0 | 15.0 | 35.6 |
| *Fine-Tuning* | Qwen2.5VL-7B PAE | 40.0 | 21.6 | 5.0 | 22.7 |
|  | Qwen2.5VL-7B SAGE | 42.0 | 27.2 | 9.0 | 27.1 |
|  | Qwen2.5VL-32B PAE | 48.0 | 32.6 | 9.0 | 31.5 |
|  | Qwen2.5VL-32B SAGE | 52.0 | 43.6 | 15.0 | 39.7 |

Table 3: **Success rate comparisons by task difficulty.** Tasks are grouped into *Easy* (<7 steps), *Medium* (7–15 steps), and *Hard* (>15 steps), covering 25%, 54%, and 21% of the WebArena task set, respectively. SAGE consistently improves performance across all difficulty levels, with especially strong gains on harder, long-horizon tasks.

low-level strategy—in enabling agents to handle complex, multi-step reasoning tasks that are typically beyond the reach of imitation-based fine-tuning.

## 5.4 Ablation Studies

In this section, we conduct several ablation studies to show the effectiveness of different components of our SAGE .

**Impact of the Top-Level Exploration Strategy.** To assess the importance of our self-adaptive task proposer, we conduct an ablation study in which it is replaced by a static task proposer that samples from a fixed distribution derived solely from the pre-exploration phase. In this setting, the agent receives tasks drawn from the same initial distribution across all training iterations, without any adjustment based on its evolving performance. While this ablated version still benefits from task diversity due to the breadth of information collected during pre-exploration, it lacks the ability to gradually increase task complexity or respond to the agent's learning progress.

As shown in Table 4 , removing the self-adaptive mechanism results in a substantial drop in performance. This highlights the importance of dynamic task generation in driving curriculum progression. The ability to promote simpler tasks when the agent struggles and introduce more challenging ones as it improves proves essential for effective skill acquisition in complex web environments.

| Model | Map | Reddit | OSS | Gitlab | CMS | Avg |
|---|---|---|---|---|---|---|
| Qwen 2.5VL-7B SAGE w/o top | 25.6 | 25.3 | 40.1 | 17.5 | 14.2 | 24.5 |
| Qwen 2.5VL-7B SAGE | 28.1 | 29.7 | 43.2 | 18.9 | 15.6 | 27.1 |
| Qwen 2.5VL-32B SAGE w/o top | 40.5 | 37.1 | 42.9 | 28.6 | 28.4 | 35.5 |
| Qwen 2.5VL-32B SAGE | 42.3 | 43.7 | 46.1 | 31.4 | 35.2 | 39.7 |

Table 4: **Ablation study on the top-level exploration strategy.** We compare performance with and without the self-adaptive task proposer on WebArena. The results show that removing top-level exploration—i.e., replacing the adaptive task curriculum with a static task distribution—leads to a consistent drop in performance across all domains and model sizes. This highlights the importance of self-adaptive task generation in driving skill acquisition.

**Impact of the Low-Level Exploration Strategy.** We further evaluate the contribution of our low-level exploration strategy by replacing both the MCTS-based rollout and the step-wise learning algorithm with a conventional learning pipeline. In this ablation, the agent samples task instructions uniformly from the task set and performs standard rollouts without search-based planning. The collected trajectories are then filtered based on task success, and a behavior cloning objective is applied to fine-tune the agent using only successful trajectories.

As shown in Table 5, removing the low-level exploration module leads to a clear drop in both sample efficiency and final task success rate. This result underscores the importance of planning and structured trajectory selection in sparse-reward environments. The MCTS-guided rollout encourages the agent to actively search for viable

| Model | Map | Reddit | OSS | Gitlab | CMS | Avg |
|---|---|---|---|---|---|---|
| Qwen 2.5VL-7B SAGE w/o low | 27.3 | 26.3 | 42.5 | 15.2 | 14.6 | 25.2 |
| Qwen 2.5VL-7B SAGE | 28.1 | 29.7 | 43.2 | 18.9 | 15.6 | 27.1 |
| Qwen 2.5VL-32B SAGE w/o low | 41.0 | 42.3 | 44.8 | 27.1 | 29.3 | 36.9 |
| Qwen 2.5VL-32B SAGE | 42.3 | 43.7 | 46.1 | 31.4 | 35.2 | 39.7 |

Table 5: **Ablation study on the low-level exploration strategy.** Removing the MCTS-based rollout and step-wise learning leads to consistent performance drops, highlighting their importance for efficient planning and long-horizon reasoning.

action sequences, whereas the normal rollout strategy often causes the agent to get stuck repeating incorrect behaviors. By enabling targeted exploration and better credit assignment, the low-level exploration module substantially enhances performance, particularly on long-horizon and more complex tasks.

**Accuracy of the Training-Based Evaluator.** To evaluate the effectiveness of our learned outcome evaluator, we compare the evaluation accuracy across different models before and after training. We first collect a set of trajectories by rolling out Qwen2.5VL-32B SAGE on held-out tasks, and then manually annotate the success or failure of each trajectory using human judgment. These human-annotated outcomes serve as ground-truth labels for evaluating different evaluator models. In our setup, we fine-tune Qwen2.5VL-7B to serve as the learned evaluator, trained using both model-generated and human-verified supervision signals.

As shown in Table 6, we observe that proprietary models initially outperform open-source models in evaluator accuracy. However, after training, our Qwen2.5VL-7B evaluator achieves the highest accuracy overall. Notably, the trained evaluator is significantly more precise in handling challenging cases, such as recognizing task infeasibility, which are critical for providing reliable supervision during agent training. These results highlight the importance of a robust, adaptive evaluator in enabling self-supervised learning.

## 5.5 Error Analysis

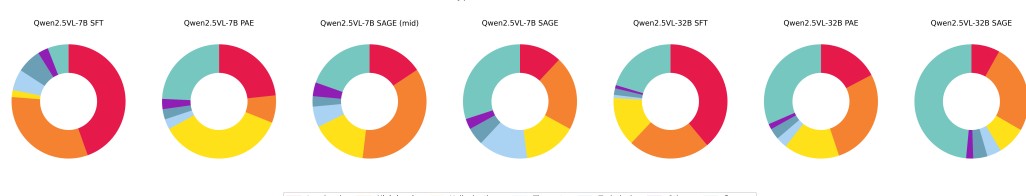

Figure 2: **Error type distribution across models.** Each donut chart illustrates the proportions of different error types and success cases on WebVoyager tasks, annotated for each model variant. SAGE effectively reduces both low-level and high-level error rates across model sizes. For Qwen2.5VL-7B, we also include an intermediate checkpoint ("mid") which shows a significant reduction in low-level errors early in training. As training progresses, high-level reasoning errors also diminish, suggesting that *SAGE enables agents to first master simple skills before tackling harder reasoning challenges*. On Qwen2.5VL-32B, SAGE achieves the lowest error rates and highest success rate, demonstrating its robustness at scale.

To better understand how SAGE facilitates skill acquisition from easy to hard tasks, we conduct a detailed error analysis on the WebVoyager [5] benchmark. We randomly sample 200 tasks from the test set and evaluate multiple models, and then manually annotate the resulting trajectories based on failure modes.

|  | Claude 3.5 Sonnet | Qwen 2.5VL-7B | SAGE |
|---|---|---|---|
| Accuracy | 82% | 75% | 90% |

Table 6: **Evaluator accuracy comparison.** Accuracy measured against human labels. We fine-tune Qwen2.5VL-7B as the learned evaluator.

Following the error types introduced in prior work [17], we categorize each failure into six distinct types:

- **Low-level execution errors:** The agent has a plausible plan but fails to interact correctly with the interface, such as clicking the wrong element or failing to navigate to the intended page.

- **High-level reasoning or planning errors:** The agent fails to formulate an effective strategy or to reason correctly through the interface in order to solve the task.

- **Visual hallucinations:** The agent fabricates answers that are not supported by the visible content, e.g., referencing information that is not present or misidentifying page content.

- **Timeouts:** The agent is on the correct path but exceeds the maximum number of interaction steps before completing the task.

- **Technical issues:** Errors resulting from environment-level problems such as broken links, loading failures, or website outages, not attributable to the agent itself.

- **Other:** Miscellaneous cases that do not fall into the above categories, including fundamentally infeasible tasks.

# 6 Conclusion

We present a new self-guided training framework for generalist foundation model web agents, centered around a multi-level hierachical exploration strategy. By decomposing the learning process into three functional phases: pre-exploration for environment understanding, top-level exploration for curriculum-driven task generation, and low-level exploration for efficient interaction—the agent is able to autonomously acquire and refine skills directly through interaction with real-world websites. Our framework eliminates the need for static task datasets or expert demonstrations, instead allowing the agent to generate its own tasks, evaluate its progress, and improve over time. Empirical results demonstrate that this self-guided training approach leads to more scalable and robust skill acquisition, enabling agents to solve complex web tasks with stronger generalization and higher efficiency. We hope this work takes a meaningful step toward the development of foundation model agents that learn from experience and adapt to diverse digital environments autonomously.

# Acknowledgments

This work was supported in part by the Amazon-Illinois Center on AI for Interactive Conversational Experiences, NSF under Grants 2106825 and 2519216, the DARPA Young Faculty Award, and the ONR Grant N00014-26-1-2099. This work used computational resources, including Amazon Web Services (AWS), the NCSA Delta and DeltaAI supercomputers through allocation CIS230012 from the Advanced Cyberinfrastructure Coordination Ecosystem: Services & Support (ACCESS) program, as well as OpenAI API through the National Artificial Intelligence Research Resource (NAIRR) Pilot.

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

# A  Algorithm Details

In this section, we provide additional details on the core algorithmic components of SAGE. We first present the Monte Carlo Tree Search (MCTS) rollout procedure used during low-level exploration, followed by the evaluator training setup that enables robust supervision without relying on ground-truth rewards.

## A.1  MCTS Rollout Procedure

Our MCTS-based rollout strategy is inspired by prior work on structured decision-making and planning [32, 33], with key adaptations for the web-based setting and multimodal agent interaction. The search tree is initialized with a single root node representing the starting state of the task, typically corresponding to the homepage URL.

**Trainable and fixed components.**  During training, the **task generator** and **evaluator** models remain fixed and only adjust their contextual inputs dynamically based on the agent's recent performance. The **agent model** is the only component optimized with gradient-based updates, while **MCTS** functions purely as a planning and sampling mechanism to collect high-quality trajectories and is not directly optimized. This design ensures stable and reproducible learning while allowing adaptive task progression.

**Training objective.**  Since only the agent is trainable, we use a step-level Direct Preference Optimization objective. Given a task prompt $x$ and two trajectories $\tau_{\text{win}}$ and $\tau_{\text{lose}}$ sharing the same prefix $(s_1, \ldots, s_k)$, where one succeeds and the other fails, the loss is:

$$L(\theta) = -\mathbb{E}_{(x, s_{1:k}, \tau_{\text{win}}, \tau_{\text{lose}}) \sim D}\Big[ \log \sigma\big(\beta\big[ \log \frac{\pi_\theta(\tau_{\text{win}} \mid x, s_{1:k})}{\pi_{\text{ref}}(\tau_{\text{win}} \mid x, s_{1:k})} - \log \frac{\pi_\theta(\tau_{\text{lose}} \mid x, s_{1:k})}{\pi_{\text{ref}}(\tau_{\text{lose}} \mid x, s_{1:k})}\big]\big)\Big],$$

where $\pi_\theta$ denotes the agent policy, $\pi_{\text{ref}}$ a frozen reference model, and $\beta$ a temperature coefficient. This objective encourages the agent to prefer trajectories leading to successful completions.

**Training procedure.**  Before the first epoch, we conduct a **pre-exploration phase** to map website structures and collect structural affordances. In each epoch, the task generator produces a batch of tasks conditioned on these structures. The agent performs MCTS-guided rollouts to explore both successful and failed trajectories. After collecting sufficient data, the agent policy is updated via the step-level DPO loss above. Task-level success statistics are then fed back into the task generator context, allowing adaptive adjustment of task difficulty for the next epoch.

**MCTS rollout.**  Each MCTS iteration proceeds through four phases: *selection*, *expansion*, *simulation*, and *backpropagation*.

**Selection.** Starting from the root, the agent traverses the tree using the Upper Confidence Bound (UCB) score [34]:

$$\text{UCB}(u) = v_u + \epsilon\sqrt{\frac{\ln n_{\text{parent}}}{n_u}},$$

where $v_u$ is the value estimate of node $u$, $n_u$ its visit count, and $n_{\text{parent}}$ the visit count of its parent node. The coefficient $\epsilon$ balances exploration and exploitation.

**Expansion.** Upon reaching a leaf node, the agent expands it by sampling multiple candidate actions from $\pi_\theta(Q, s)$, each producing a new child state. The value of each child $v_{s'}$ is initialized using the evaluator to estimate task success likelihood, providing a prior for downstream planning.

**Simulation.** From the most promising child node, a rollout is performed up to a fixed depth $L$. At each step, the agent greedily selects the top-confidence action. The trajectory terminates upon task completion or reaching the step limit.

**Backpropagation.** After the rollout, a binary success reward is propagated upward to update value estimates and visit counts. The MCTS process continues for a fixed number of iterations.

**Algorithm 1** Low-Level Exploration via MCTS

---

**Input:** Task $Q$, agent policy $\pi$, initial page $p_0$, value model $V$, max iterations $N$, episode length $L$
**Initialize:** $T \leftarrow$ InitializeTree($p_0$)

1:  **for** $i = 1$ to $N$ **do**
2:      $s \leftarrow \text{Root}(T)$
3:      **while** $s$ is not a leaf node **do**
4:          $s \leftarrow \arg\max_{s'} \left( v_{s'} + \epsilon \sqrt{\frac{\log n_s}{n_{s'}}} \right)$
5:      **end while**
6:      **if** $s$ is not terminal **then**
7:          $a_{1:k} \leftarrow \pi(Q, s)$
8:          $s'_{1:k} \leftarrow \text{ApplyActions}(s, a_{1:k})$
9:          $v_{s'_j} \leftarrow V(s'_j)$ for all $j$
10:          $s' \leftarrow \arg\max_j v_{s'_j}$
11:          $r \leftarrow \text{RolloutFrom}(s', \text{depth} = L - \text{depth}(s'))$
12:          UpdateNode($s'$, reward $= r$)
13:      **end if**
14:      BackPropagate($s$)
15: **end for**

---

**Low-level exploration pseudocode.**    For clarity, we summarize the low-level MCTS exploration procedure below.

This integration of MCTS planning with gradient-based DPO training enables SAGE to explore promising trajectories more efficiently and improve success rates across complex, long-horizon web tasks.

## A.2    Confidence Prediction Training.

The agent policy $\pi_\theta$ is augmented with a scalar confidence head that outputs $c \in [0, 1]$ for each action. For initialization, we prompt the proprietary model Claude Sonnet 3.5 to generate demonstration trajectories where each action is annotated with a self-estimated confidence score. These annotated trajectories are used during supervised fine-tuning to initialize the confidence prediction capability of our base model.

During the subsequent SAGE training phase, the agent's confidence estimates are further refined within the step-level DPO framework. Since the MCTS-based sampling process does not always follow the action with the highest predicted confidence, a realignment step is applied to calibrate confidence with actual success outcomes. Specifically, in successful trajectories, the executed action is assigned a confidence value of 1, ensuring that the model learns to associate high confidence with decisions that lead to successful task completion. This refinement improves the calibration between model certainty and real-world decision reliability, yielding more stable long-horizon exploration.

## A.3    Evaluator Training

To enable reliable supervision without access to ground-truth reward functions, we train a task outcome evaluator using the open-source model Qwen2.5VL-7B [35]. The evaluator is designed to predict task success or failure based on evidence collected during agent interaction. Its input consists of: (1) the task instruction, (2) the final answer generated by the agent, (3) the full interaction trajectory, and (4) multimodal evidence composed of the final three screenshots and their associated accessibility tree. The output is a binary success label, optionally accompanied by a natural language explanation.

We construct the training dataset in two stages. First, we bootstrap the evaluator using over 10,000 evaluation traces generated by Claude 3.5 Sonnet [29], treating its responses as pseudo-labels. These include both reasoning and binary outcome labels. This initial phase enables the evaluator to inherit strong generalization capabilities from the proprietary model.

In the second stage, we refine the evaluator using a smaller, high-quality human-annotated dataset. Specifically, we randomly sample 1,200 trajectories from the initial dataset and manually annotate their ground-truth success labels. Among these, we identify approximately 400 instances where Claude's predictions disagree with human judgment. For each of these error cases, we provide a corrected binary label and, when relevant, a brief explanation describing the evaluation mistake. The format remains consistent with the initial dataset, enabling seamless fine-tuning. This two-stage process enhances the evaluator's precision and robustness, particularly for nuanced or failure-prone cases.

## B  Environment Details

In this section, we describe the multimodal web environments used in our experiments, including implementation details, observation formatting, and action space definitions.

**Environment Overview.**  Our environment is implemented using a combination of Selenium[1] and Playwright[2], and wrapped into a standardized Gym interface to support reinforcement learning workflows. In addition to basic functionalities required for agent-environment interaction, we extend the framework with support for saving and restoring environment states at arbitrary time steps. This feature is critical for enabling tree-based search strategies such as MCTS, which require repeated rollouts from intermediate checkpoints.

**Observation Details.**  As illustrated in Figure 3, each observation consists of two aligned modalities: a rendered screenshot with overlaid set-of-marks, and a structured accessibility tree. In our implementation, we first extract the set-of-marks and the accessibility tree independently. We then compute a matching between elements in the two modalities and align the numeric labels in the set-of-marks with the corresponding nodes in the accessibility tree. This alignment ensures that agents can refer to web elements consistently across visual and structural representations.

**Action Space.**  Following the setup in PAE [17], we define a discrete action space that includes the following interaction types:

- **Click:** Click on a labeled element such as a button or link.
- **Type:** Enter text into an input field.
- **Scroll:** Scroll a scrollable container or the entire page.
- **Return:** Navigate back to the previous page in browser history.
- **Answer:** Return the final answer.

## C  Additional Experiments

In this section, we present supplementary experimental results to further analyze the effectiveness of individual components in our framework.

### C.1  Ablation Study: Pre-Exploration Strategy

To evaluate the impact of the pre-exploration strategy, we conduct an ablation experiment where this phase is removed. In this variant, the task generator only receives the base URL of the website along with trajectories collected from previous training iterations. Without the structured understanding provided by pre-exploration, the generator lacks information about the page layout and interaction flow.

As shown in Table 7, removing pre-exploration leads to a noticeable degradation in performance. The task generator in this setting frequently proposes irrelevant or infeasible tasks due to its limited knowledge of website structure, demonstrating the importance of pre-exploration in grounding curriculum design.

---

[1]https://pypi.org/project/selenium/
[2]https://playwright.dev/

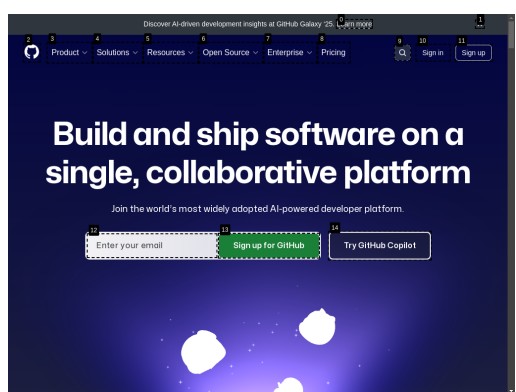

```
[333] link 'Skip to content'
[360] alert 'Announcement'
  [0] link 'Learn more'
  [1] button 'Close'
[387] heading 'Navigation Menu'
  [2] link 'Homepage'
[434] navigation 'Global'
  [3] button 'Product'
  [4] button 'Solutions'
  [5] button 'Resources'
  [6] button 'Open Source'
  [7] button 'Enterprise'
  [8] link 'Pricing'
  [9] button 'Search or jump to...'
 [10] link 'Sign in'
 [11] link 'Sign up'
```

Figure 3: **Multimodal environment input.** Left: Screenshot of a rendered webpage with set-of-marks. Right: Accessibility tree snippet showing the hierarchical structure of interactive elements. This multimodal input enables agents to understand both visual and structural content for accurate interaction. The agent can find buttons both from the vision input and the text input.

## C.2 Ablation Study: Evaluator

We conduct an ablation study to assess the importance of the training-based evaluator in our framework. Specifically, we replace the learned evaluator with alternative models—Claude 3.5 Sonnet and Qwen2.5VL-32B—and use their raw outputs to supervise agent training. As shown in Table 8, our training-based evaluator leads to better task success rates than both baselines, *despite being based on a smaller open-source model*. This highlights the benefit of targeted evaluator fine-tuning on trajectory outcomes, which enhances the quality and reliability of training supervision.

We present a detailed error breakdown in Figure 2 to examine how SAGE facilitates progressive skill acquisition across training stages and model scales. Across all models, we find that SAGE significantly improves task performance by reducing two major sources of failure: low-level execution errors and high-level planning errors. Notably, in the Qwen2.5VL-7B model, we analyze an intermediate checkpoint after 5 training iterations and observe a marked decrease in low-level failures—indicating that the agent learns precise actions early on in the training process, largely through

| Model | Map | Reddit | OSS | Gitlab | CMS | Avg |
|---|---|---|---|---|---|---|
| Qwen 2.5VL-7B SAGE w/o pre | 20.3 | 12.4 | 26.6 | 9.3 | 8.8 | 15.3 |
| Qwen 2.5VL-7B SAGE | 28.1 | 29.7 | 43.2 | 18.9 | 15.6 | 27.1 |
| Qwen 2.5VL-32B SAGE w/o pre | 40.3 | 24.1 | 32.0 | 18.3 | 20.8 | 27.1 |
| Qwen 2.5VL-32B SAGE | 42.3 | 43.7 | 46.1 | 31.4 | 35.2 | 39.7 |

Table 7: **Ablation study on the pre-exploration strategy.** We evaluate the impact of removing the pre-exploration phase, where the task generator receives no structural information about the website beforehand. Without this step, the task generator becomes less grounded and often infeasible, resulting in a significant performance drop across all categories. These results demonstrate the importance of pre-exploration for understanding environment structure and supporting meaningful task generation.

| Model | Map | Reddit | OSS | Gitlab | CMS | Avg |
|---|---|---|---|---|---|---|
| Qwen 2.5VL-7B SAGE Claude | 24.3 | 25.3 | 38.4 | 16.3 | 14.2 | 23.0 |
| Qwen 2.5VL-7B SAGE Qwen | 23.2 | 26.3 | 36.2 | 16.0 | 14.6 | 22.7 |
| Qwen 2.5VL-7B SAGE | 28.1 | 29.7 | 43.2 | 18.9 | 15.6 | 27.1 |
| Qwen 2.5VL-32B SAGE Claude | 41.7 | 40.8 | 41.3 | 29.6 | 32.4 | 36.4 |
| Qwen 2.5VL-32B SAGE Qwen | 39.9 | 40.7 | 41.6.0 | 25.8 | 30.2 | 34.5 |
| Qwen 2.5VL-32B SAGE | 42.3 | 43.7 | 46.1 | 31.4 | 35.2 | 39.7 |

Table 8: **Ablation study on the evaluator module.** We compare our training-based evaluator against two alternatives: Claude 3.5 Sonnet and Qwen2.5VL-32B, both used as fixed evaluators without additional fine-tuning. Despite its smaller scale, our trained evaluator consistently yields higher task success rates, demonstrating the importance of domain-adapted supervision for reliable reward estimation.

practicing easier tasks. As training progresses, the proportion of high-level planning errors decreases as well, reflecting the agent's transition to mastering more complex tasks. This learning trajectory aligns with the curriculum induced by our top-level exploration strategy. Furthermore, on the larger Qwen2.5VL-32B model, SAGE consistently reduces both types of errors and achieves the highest success rate among all variants, highlighting its scalability and effectiveness even with stronger backbones.

## C.3 Evaluator Fine-tuning and Cross-Framework Comparison

We further analyze the role of limited human annotations in SAGE. Only a small amount of human-labeled data (less than eight hours of effort by one author) is used for fine-tuning the outcome evaluator in the final stage, while all other components are trained fully autonomously. This small supervision improves the evaluator's accuracy but has minimal effect on overall system performance—SAGE's main gains stem from its adaptive curriculum and exploration strategies. The comparison in Table 9 also shows that even when PAE

| Model | Map | Reddit | OSS | Gitlab | CMS | Avg |
|---|---|---|---|---|---|---|
| Qwen2.5VL-7B PAE | 23.8 | 24.6 | 37.4 | 15.3 | 12.4 | 22.7 |
| Qwen2.5VL-7B PAE + SAGE 's Eval | 24.1 | 24.4 | 37.6 | 14.9 | 12.9 | 22.9 |
| Qwen2.5VL-7B SAGE | 28.1 | 29.7 | 43.2 | 18.9 | 15.6 | 27.1 |
| Qwen2.5VL-32B PAE | 34.5 | 35.9 | 41.0 | 24.1 | 22.0 | 31.5 |
| Qwen2.5VL-32B PAE + SAGE 's Eval | 35.8 | 36.2 | 43.1 | 26.4 | 23.2 | 33.0 |
| Qwen2.5VL-32B SAGE | 42.3 | 43.7 | 46.1 | 31.4 | 35.2 | 39.7 |

Table 9: **Evaluator fine-tuning and cross-framework comparison.** We compare PAE and SAGE under identical evaluators to isolate the effect of evaluator quality. Even when using the same trained evaluator, SAGE substantially outperforms PAE, confirming that the main performance gain arises from our hierarchical exploration and curriculum mechanism rather than evaluator differences.

adopts the same trained evaluator, the performance gap remains large, underscoring that SAGE effectively leverages stronger evaluators through its hierarchical design.

## C.4 Efficiency of Low-Level Exploration

To quantify the efficiency of our low-level exploration strategy, we compare the number of successful trajectories generated under the same sampling budget of 10k trajectories per model. As shown in Table 10, the low-level exploration consistently produces a higher proportion of successful rollouts, demonstrating improved sample efficiency and faster policy improvement over standard rollouts.

| Model | # Successful Trajectories (10k Samples) |
|---|---|
| Qwen2.5VL-7B + Normal | 1937 |
| Qwen2.5VL-7B + Low-level Exploration | 2102 |
| Qwen2.5VL-32B + Normal | 2578 |
| Qwen2.5VL-32B + Low-level Exploration | 2895 |

Table 10: **Sample efficiency of low-level exploration.** We compare the number of successful trajectories collected under identical sampling budgets. The proposed low-level MCTS-based exploration consistently yields more successful rollouts than standard rollouts, confirming its superior sample efficiency and stronger training signal.

## C.5 SFT Baseline Comparison

For completeness, we report the performance of the supervised fine-tuned (SFT) baselines that serve as initialization for both PAE and SAGE. As shown in Table 11, SAGE consistently achieves higher success rates than both SFT and PAE across all benchmarks, further validating the effectiveness of our hierarchical training paradigm.

| Model | Map | Reddit | OSS | Gitlab | CMS | Avg |
|---|---|---|---|---|---|---|
| Qwen2.5VL-7B SFT | 16.5 | 21.3 | 32.6 | 13.7 | 10.1 | 19.4 |
| Qwen2.5VL-7B PAE | 23.8 | 24.6 | 37.4 | 15.3 | 12.4 | 22.7 |
| Qwen2.5VL-7B SAGE | 28.1 | 29.7 | 43.2 | 18.9 | 15.6 | 27.1 |
| Qwen2.5VL-32B SFT | 27.3 | 27.8 | 38.0 | 18.2 | 14.7 | 25.7 |
| Qwen2.5VL-32B PAE | 34.5 | 35.9 | 41.0 | 24.1 | 22.0 | 31.5 |
| Qwen2.5VL-32B SAGE | 42.3 | 43.7 | 46.1 | 31.4 | 35.2 | 39.7 |

Table 11: **Comparison with SFT baselines.** We report results of the supervised fine-tuned (SFT) models for completeness. SAGE consistently outperforms both SFT and PAE across all environments and model scales, confirming the benefit of hierarchical exploration over static imitation learning.

## D Hyperparameters

We include the hyperparameters that we have used in Table 12.

## E Prompts

We include all prompting templates used throughout our framework. In particular, Figures 5 and 6 show the prompt used during the *pre-exploration phase*, where the agent is instructed to explore and describe the structure and functionality of a given website starting from its homepage. Figures 7 and 8 provide the prompt template used in the *top-level exploration phase*, where the task generator proposes new instructional tasks based on previous trajectories and structural understanding of the

| Hyperparameter | Value |
| --- | --- |
| number of actions | 10 |
| maximum exploration step | 100 |
| number of tasks pre-generated | 10, 000 |
| number of tasks generated in iteration | 2, 000 |
| number of iterations in MCTS | 25 |
| number of trajectories | 2, 048 |
| actor update epochs per iteration | 4 |
| batch size | 8 |
| DPO $\beta$ | 0.45 |

Table 12: Hyperparameters used for SAGE.

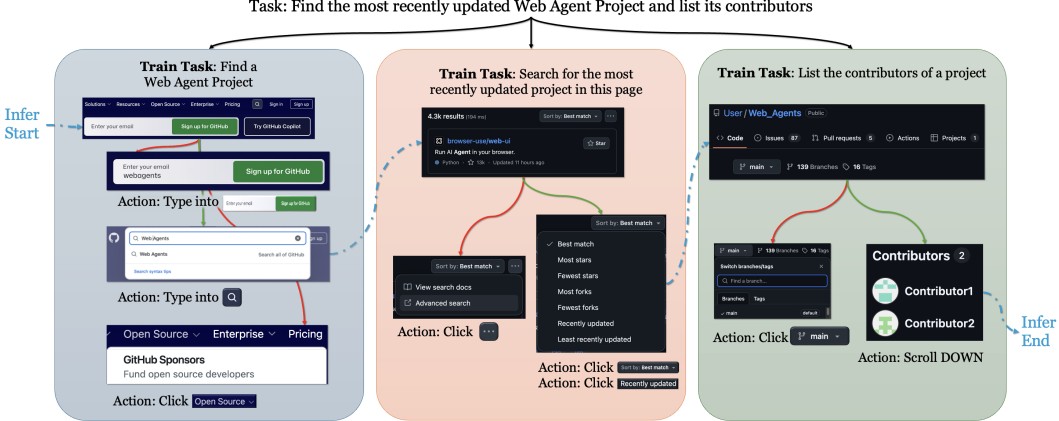

Figure 4: An example showing how SAGE facilitates skill acquisition through hierarchical exploration. The agent initially fails to complete the task "Find the most recently updated Web Agent project and list its contributors." In subsequent iterations, the task generator offers simpler subtasks with the *top-level exploration*. The agent learns to solve each subtask individually through the *low-level exploration*, ultimately enabling it to succeed on the original, more complex task. The blue line shows a trajectory the agent generated during the inference phase.

environment. All prompts are designed to be general and domain-agnostic, ensuring compatibility across both open-source and proprietary models.

# F    A Case Study of SAGE

We provide a case study that illustrates how SAGE enables an agent to acquire complex skills through structured, hierarchical exploration. We examine a challenging task from the GitHub website: *"Find the most recently updated Web Agent project and list its contributors."* As shown in Figure 4, prior to training, the agent consistently fails to initiate the correct actions, struggling even to identify how to search for relevant projects.

Through the *top-level exploration* module, SAGE's self-guided task generator decomposes the difficult instruction into three simpler subtasks: (1) locating a Web Agent project, (2) identifying the most recently updated entry, and (3) retrieving its contributors. These tasks are short-horizon and require only a few well-defined interactions, allowing the agent to explore and acquire each skill individually via *low-level exploration*.

Once the agent demonstrates competence on the subtasks, the task generator reintroduces the original complex task, allowing the agent to practice solving it end-to-end. Empowered by prior experience and learned sub-skills, the agent successfully completes the original instruction—demonstrating the effectiveness of SAGE in enabling curriculum-driven skill composition and long-horizon reasoning.

# G  Limitations and Future Work

While SAGE achieves strong performance across diverse web environments, it is not without limitations. First, our framework assumes access to a stable and scriptable browser environment, which may not hold in fully open-web scenarios that include CAPTCHAs, login requirements, or highly dynamic JavaScript content. Although we use Playwright wrappers to store and reload webpage states—enabling reproducible resets similar to prior works [17**?** ]—this assumption limits the applicability of our MCTS-based exploration in real-world settings where actions may be irreversible or pages non-deterministic. Future extensions may incorporate conservative planning, safe policy filtering, or human-in-the-loop oversight to enable safer deployment in uncontrolled web environments.

Second, SAGE introduces several environment-specific augmentations—such as leveraging webpage accessibility trees, BFS-based initial trajectory collection, and full environment resets—that simplify training in standardized benchmarks but may not generalize to arbitrary or uninstrumented websites. Improving robustness under weaker assumptions (e.g., partial observability or limited environment control) remains an important avenue for future research.

Third, our current experiments are conducted in standardized, resettable environments that require substantial computational resources for large-scale MCTS rollouts and high-fidelity webpage rendering. This high barrier to replication may hinder broader community adoption or re-implementation without comparable infrastructure. We plan to mitigate this by releasing lightweight simulation setups and environment wrappers to facilitate partial reproduction under modest compute budgets.

Finally, although SAGE is designed for research and sandboxed environments, applying it directly to live websites may raise privacy concerns, as agents could unintentionally access or leak user-sensitive information during open-ended exploration. Ensuring privacy-safe deployment and extending our approach beyond web navigation—e.g., to desktop or mobile human-computer interaction—represent key directions for future work.

# H  Broader Impact

This work aims to advance the development of autonomous web agents that can interact with websites using vision-language foundation models. Such systems have the potential to automate a wide range of digital tasks—ranging from information retrieval to workflow assistance—thereby improving accessibility and efficiency for users. At the same time, their interaction with real-world websites introduces important safety and ethical considerations.

**Controlled research setting.**  All experiments in SAGE are conducted within strictly controlled environments. WebArena is a fully sandboxed simulator, and WebVoyager provides access only to publicly available webpages without user authentication or private data. All web interactions are executed inside a secure, firewall-protected network that monitors and blocks potentially unsafe requests. The agent is never provided with personal information, and all tasks involving user accounts or sensitive data are explicitly excluded. These measures ensure that SAGE operates entirely within a research context and cannot cause real-world effects.

**Safety and misuse considerations.**  We recognize that autonomous web agents raise broader safety risks, including unintended interactions with real systems, information leakage, or malicious misuse. Although such risks are absent in our current setup, future large-scale deployments will require robust safeguards, such as: (1) decoupling policy outputs from execution through a controlled intermediary layer with safety filters, (2) adding dedicated safety-monitor models to detect and block risky actions, (3) enforcing rate limiting, permission gating, and anomaly detection, and (4) maintaining human-in-the-loop oversight during execution. We emphasize that our work focuses on algorithmic capability under safe, offline environments, but we strongly support the establishment of community standards for safe deployment of web agents.

**Ethical and privacy implications.**  Interactions with real-world websites can raise privacy or data protection concerns. While our experiments do not involve any private or user-generated content, future work must ensure that autonomous web agents respect consent boundaries and data-use policies. We also acknowledge potential social risks, such as unintended automation of sensitive or

economically impactful tasks. These issues call for broader interdisciplinary research that combines technical safeguards with ethical and policy frameworks to ensure responsible deployment.

**Outlook.** Our primary goal is to improve the technical understanding and safe design of autonomous web agents. We believe that future progress in this field must go hand in hand with advances in safety, interpretability, and accountability mechanisms. We are committed to contributing to the development of transparent, privacy-conscious, and human-aligned web agent systems that can benefit society while minimizing risk.

# I  Ethical and Safety Considerations

This work develops SAGE as a research framework for studying autonomous web agents in controlled environments. All experiments are conducted entirely within **WebArena** and **WebVoyager**, following prior work [17]. **WebArena** is a fully synthetic sandbox that simulates websites locally without any network connection or real data. **WebVoyager** contains only a small set of manually curated, static webpages from publicly accessible domains (e.g., Google Maps, Wikipedia), without user authentication, private data, or executable operations. Agent activities occur behind a firewall, and all interactions are logged for analysis and accountability. Consequently, SAGE does not access or collect any sensitive or personalized information.

We acknowledge the broader ethical and security concerns associated with deploying fully autonomous agents on the open web. Although SAGE poses no privacy or misuse risk in its current form, future applications should incorporate additional safeguards such as restricted execution privileges, rate limiting, security monitoring, and human-in-the-loop approval before critical actions. Detailed action logs and auditing mechanisms should also be used to ensure accountability and prevent unintended behaviors.

Finally, no crowd-sourced or external annotation was used in this project. All labeled examples for evaluator calibration were created internally by the authors. We oppose any unauthorized or adversarial use of this system and encourage future research to adhere to responsible, transparent, and privacy-preserving development practices.

**Task Generator Prompt for SAGE (Part A)**

{"web_name": "Amazon",

"web": "https://www.amazon.com/",

"structure_info": ["Homepage includes sections such as 'Electronics', 'Books', and 'Deals'.", "Product pages contain price, reviews, delivery info, and variants.", "Search results are filterable by price, rating, and brand."],

"previous_trajectories": [

{"instruction": "Search for headphones under $50 with at least 4 stars.", "success rate": 0.32},

{"instruction": "Compare features between AirPods Pro and Galaxy Buds.", "success rate": 0.00},

{"instruction": "Find a gift card worth exactly $100.", "success rate": 0.90}

]}

We are training a web agent that learns to complete real-world tasks by interacting with web interfaces. Your job is to generate high-quality, instructional tasks that help the agent improve over time. The agent has already explored this website and attempted previous tasks. Based on this prior experience and structural understanding of the site, you must generate a new batch of 25 training instructions.

**Step 1: Reasoning Instructions**

Before listing your output, write your reasoning on how you adapted your tasks. Your strategy must consider the following:

- **Simplify low-success tasks:** For instructions with low success (e.g., below 40%), generate 1–2 simpler variants by relaxing filters or breaking down subtasks.

- **Harder from high-success tasks:** For instructions with high success (e.g., above 80%), generate harder variants by adding filters or composing with other subtasks.

Figure 5: Task Generator Prompt Part A.

**Task Generator Prompt for SAGE (Part B)**

- **Structure-aware task design:** Use the provided `structure_info` to ensure realism.

- **Curriculum balance:** Ensure a mix of easy, medium, and hard tasks.

**Step 2: Output Format**

- Output exactly **25 instructions** in **JSONL format**.

- Each line must be a valid JSON object with a single field `"instruction"`.

**Rules and Constraints**

- Do not require login, user data, or current time.

- Ensure task is verifiable via screenshots.

- Completion should require 2–8 steps.

**In-Context Examples:**

- *Simplified:* {"instruction": "Search for headphones under $50."}

- *Harder:* {"instruction": "Compare Galaxy Buds and AirPods, then find a cheaper rated alternative."}

Figure 6: Task Generator Prompt Part B.

**Pre-Exploration Prompt (Part A)**

You are an intelligent web browsing agent designed to explore and understand new websites before learning specific tasks. In this phase, you will only receive the homepage of a new website, and your goal is to discover and record meaningful pages and their functions. This phase is unguided—you are not solving any tasks.

In each step:

- You will observe a screenshot, accessibility tree, and numerical element labels.

- Recall visited pages to avoid redundant navigation.

- Propose up to **10 distinct actions** that could lead to semantically different states.

- Record what the page reveals (e.g., filters, product listings, FAQs).

Figure 7: Pre-Exploration Prompt Part A: Instructions for agent behavior and high-level goals.

---

**Pre-Exploration Prompt (Part B)**

**Output Format per step:**

- **Thought:** Reason about the state and exploration plan.

- **Top-10 Actions:** Propose 10 diverse actions (e.g., Click [3], Scroll WINDOW; down).

- **Summary:** Describe the page's purpose or findings.

**Exploration Guidelines:**

- Avoid login, signup, or non-navigational elements.

- Maximize diversity—explore filters, categories, products, help pages, etc.

- Use accessibility tree to guide semantic understanding.

- Avoid repeated or redundant actions.

**Example Output:**

Thought: The homepage includes categories and a search bar. I haven't explored "Books" or filtered search results yet.

Top-k Actions:

- Click [3]

- Click [4]

- Scroll WINDOW; down

- Type [10]; "usb-c hub"

- ...

Summary: This is the homepage with links to major categories and a search bar. I begin mapping product pages and filter paths.

Figure 8: Pre-Exploration Prompt Part B: Format, examples, and behavior guidelines for site exploration.

