# OpenReview forum: "Self-Guided Hierarchical Exploration for Generalist Foundation Model Web Agents"
_NeurIPS.cc/2025/Conference — NeurIPS 2025 poster_

### Official Review · Reviewer_3FCb · 2025-06-18

**Clarity:** 2
**Significance:** 3
**Originality:** 3
**Rating:** 4
**Confidence:** 3

**Summary:**

The paper describes a protocol for collecting training data for web agents by largely unsupervised self-guided exploration. The protocol is rather complex, it involves three stages. The first phase is the structural exploration of website that results in a knowledge base of accessible URLs. The second phase is adaptive task generation that takes into account the agent’s current performance. The third phase is fine grained exploration to solve as many tasks as possible using MCTS. Experiments show that the proposed protocol helps achieve higher performance compared to a strong baseline.

**Questions:**

- My understanding is that MCTS requires tight control over the state of the environment? How were able to do this in the case of the open-web WebVoyager task?
- lines 378 - 381: "removing the low-level exploration module leads to a clear drop in both sample efficiency and final task success rate" -  I did not find sample efficiency results in Table 5
- lines 297 - 300: the paper mentions SFT results with the SFT data coming from Claude; I did not find these results in the paper.
- can you talk more about the human annotation efforts in the paper? I see some information in the supplementary material, but I think it is important enough to be discussed in the main text.
- what would be SAGE's performance without human annotation?
- I think a better explanation of the baseline PAE method would improve the paper

**Ethical Concerns:**

["NO or VERY MINOR ethics concerns only"]

**Final Justification:**

I will maintain my original assessment.

**Limitations:**

How would this method work for pure JavaScript web pages that mostly use the same URL?

**Paper Formatting Concerns:**

No formatting concerns.

**Quality:**

3

**Strengths And Weaknesses:**

The main strength of the paper is that it reports very strong empirical results. Increasing the performance of Qwen32B from 49.7 to 62.4 is quite impressive. But the method is very complex and the presentation is rather high-level, it would be very hard to reproduce the results from this paper. Additionally, the paper glosses over some crucial details, such as e.g. “a limited number of human-annotated examples” that were used to train the reward model. One can see from Table 6 that the finetuned Qwen2.5VL-7B evaluator is in fact stronger than the frontier models. I’d guess it required a substantial data collection, and hence the exploration was not really purely self-guided in the presented study.

There are other things that are not very clear from the paper, see “Questions” section.

Overall, the paper's results are worth sharing, but I encourage the authors to provide more details to increase reproducibility.

---

> ### Author Rebuttal · Authors · 2025-07-31
>
> Thanks a lot for your insightful and inspiring comments! We provide the following clarifications in response to your concerns:
>
>
>
> 1. **Human-annotated data**
>
>  - We would like to clarify that our framework is largely self-guided and relies only on a very small amount of human-annotated data. Specifically, we collected 1,200 trajectories with human annotations to fine-tune the outcome evaluator in the final stage of training. These annotations do not form the **core** of the agent training pipeline.
>
>  - To demonstrate that SAGE remains effective even without any human annotations, we provide an ablation study. As shown in the table below (also refer to the table 2. In the appendix), the performance drop is minor when removing the human-annotated data, confirming that our method’s strong performance is driven by autonomous exploration rather than manual supervision:
>
> | Model                          | Map  | Reddit | OSS  | Gitlab | CMS  | Avg  |
> |--------------------------------|:----:|:------:|:----:|:------:|:----:|:----:|
> | Qwen2.5VL-7B SAGE (Claude)     | 24.3 | 25.3   | 38.4 | 16.3   | 14.2 | 23.0 |
> | Qwen2.5VL-7B SAGE (Qwen)       | 23.2 | 26.3   | 36.2 | 16.0   | 14.6 | 22.7 |
> | Qwen2.5VL-7B SAGE              | 28.1 | 29.7   | 43.2 | 18.9   | 15.6 | 27.1 |
> | Qwen2.5VL-32B SAGE (Claude)    | 41.7 | 40.8   | 41.3 | 29.6   | 32.4 | 36.4 |
> | Qwen2.5VL-32B SAGE (Qwen)      | 39.9 | 40.7   | 41.6 | 25.8   | 30.2 | 34.5 |
> | Qwen2.5VL-32B SAGE             | 42.3 | 43.7   | 46.1 | 31.4   | 35.2 | 39.7 |
>
>  - This result shows that even using raw model evaluators without human annotations, SAGE still performs competitively, and the additional fine-tuning only provides a modest improvement.
>
> 2. **MCTS in open-web environments**
>
>  - We agree with the reviewer that MCTS typically requires tight control over the environment state. In our framework, we address this by using Playwright to wrap the webpage state, allowing us to store and reload the web page. This enables the reset functionality required by MCTS. Regarding open-web WebVoyager versus the sandboxed WebArena, there is no fundamental difference in how MCTS operates: while some webpages in WebVoyager may occasionally change, they generally remain stable over the short time horizon of a single search process. If a specific branch of the search tree cannot be reliably replayed due to dynamic content, we simply discard that branch and continue the search. We acknowledge that our current approach is less robust for pure JavaScript single-page applications that heavily reuse the same URL, and we consider improving MCTS handling for such dynamic websites an interesting direction for future work.
>
> 3. **Efficiency of low-level exploration**
>
>  - We appreciate the reviewer’s question regarding the efficiency of our low-level exploration strategy. In our context, sample efficiency refers to the success trajectories achieved by the agent when generating trajectories under the same number of tasks tried. Compared to a standard rollout strategy, our low-level exploration prioritizes promising action sequences and avoids repeatedly sampling failed trajectories. This results in a higher proportion of successful trajectories for the same number of samples, which translates into a more effective training signal and faster policy improvement.
>
>  - To provide direct evidence, we conducted an additional ablation study on WebArena using the same initial SFT models. For each model, we sampled 10k trajectories using either standard rollouts or our low-level exploration. As shown in the table below, the low-level exploration strategy consistently produced more successful trajectories, which validates its superior sample efficiency:
>
>
> | Model                  | # Successful Trajectories (10k Samples) |
> |------------------------|-----------------------------------------|
> | Qwen2.5VL-7B + Normal  | 1937                                    |
> | Qwen2.5VL-7B + Low     | 2102                                    |
> | Qwen2.5VL-32B + Normal | 2578                                    |
> | Qwen2.5VL-32B + Low    | 2895                                    |
>
>  - By enabling targeted search over the action space, our low-level module improves not only the learning speed but also the final task success rates, as confirmed by our main results and ablations in Table 5.
>
>
> 4. **SFT claude data**
>
>  - Different from PAE, which emphasizes demonstrating that the proposer–agent–evaluator framework can improve performance based on SFT models, our focus is on showing that SAGE provides a more effective training paradigm compared with PAE. Due to space constraints, we did not include the SFT baseline results in the main text. For completeness, we now report the SFT performance on WebArena, compared with PAE and SAGE in the table below.
>
> | Model                | Map  | Reddit | OSS  | Gitlab | CMS  | Avg  |
> |----------------------|:----:|:------:|:----:|:------:|:----:|:----:|
> | Qwen2.5VL-7B SFT     | 16.5 | 21.3   | 32.6 | 13.7   | 10.1 | 19.4 |
> | Qwen2.5VL-7B PAE     | 23.8 | 24.6   | 37.4 | 15.3   | 12.4 | 22.7 |
> | Qwen2.5VL-7B SAGE    | **28.1** | **29.7** | **43.2** | **18.9** | **15.6** | **27.1** |
> | Qwen2.5VL-32B SFT    | 27.3 | 27.8   | 38.0 | 18.2   | 14.7 | 25.7 |
> | Qwen2.5VL-32B PAE    | 34.5 | 35.9   | 41.0 | 24.1   | 22.0 | 31.5 |
> | Qwen2.5VL-32B SAGE   | **42.3** | **43.7** | **46.1** | **31.4** | **35.2** | **39.7** |
>
> 5. **Comparison with PAE**
>
>  - The baseline method PAE establishes a general *self-improving framework* for web agents by combining three components: a proposer that generates diverse task instructions, an agent that attempts to solve these tasks, and an evaluator that judges task completion to provide rewards. This design successfully removes the reliance on manually curated datasets, enabling web agents to improve through autonomous interaction with web environments. However, PAE samples tasks from a static instruction distribution and does not explicitly address the progression from simple to complex tasks or the challenges of long-horizon exploration.
>
>  - Building on this foundation, our work introduces a *hierarchical exploration strategy* as the **core** of SAGE. Specifically, we propose 1) a self-guided task proposer that adaptively adjusts task difficulty by analyzing the agent’s performance and decomposing difficult tasks into simpler subtasks, and 2) a low-level exploration mechanism using MCTS-based planning and step-wise learning to improve trajectory quality and sample efficiency. These contributions transform the training process into a dynamic curriculum that allows agents to acquire fundamental skills first and then tackle increasingly complex tasks. As a result, SAGE significantly improves performance compared to PAE, while maintaining the fully autonomous training paradigm.
>
> 6. **Complexity and reproducibility**
>
>  - We acknowledge that SAGE involves multiple components, but each module is designed to address a specific challenge inherent to training autonomous web agents. The pre-exploration stage reduces the enormous task space by systematically mapping website structures; the top-level exploration module tackles the difficulty of long-horizon tasks by generating a self-evolving curriculum from easy to hard; and the low-level exploration module improves the agent’s ability to explore and solve complex tasks efficiently.
>
>  - To ensure reproducibility, we have already provided extensive technical details in both the main paper and the supplementary material, including full descriptions of the pre-exploration process, task generator prompts, pseudocode for the MCTS rollout, hyperparameter configurations, and evaluator training procedures. Furthermore, we will release the full codebase and trained models upon acceptance to facilitate reproducibility and further research.

---

> > ### Comment · Reviewer_3FCb · 2025-08-09
> > **thank for your response**
> >
> > Thank you for presenting additional results and answering my questions. I can see some percentage of gains went away when human data was not used. I will  maintain my score.

---

> > > ### Author Response · Authors · 2025-08-09
> > > **Response to reviewer 3FCb**
> > >
> > > Thank you for the thoughtful follow-up and for engaging in this discussion.
> > >
> > >  - First, we would like to clarify that the amount of human-annotated data used in SAGE is extremely small, which was **collected by one of the authors and required less than 8 hours of annotation time**. These annotations are solely used for fine-tuning the outcome evaluator in the final stage of training, and do not form part of the core autonomous exploration and learning process. The rest of the system operates fully autonomously without manual supervision.
> > >
> > >  - While this small amount of annotation improves the evaluator’s accuracy, the majority of SAGE’s performance gains come from our hierarchical exploration design, as evidenced by the ablation study included in the rebuttal.
> > >
> > >  - We would also like to note that, while this small amount of human-annotated data benefits SAGE’s evaluator, it may not have the same effect for PAE. Due to the limited rebuttal window, we were unable to complete the experiment combining PAE with SAGE’s evaluator trained on human-annotated data. Instead, we present results from preliminary experiments indicating PAE’s inherent limitations in exploiting a stronger evaluator. Specifically, replacing PAE’s evaluator with different pretrained models yielded only marginal performance changes – for example, with LLaVA-34B as the base agent, using a Claude-based evaluator achieved an average score of 33.0, while a Qwen-7B-based evaluator resulted in 32.7. These results suggest that **evaluator quality has limited impact on PAE’s overall performance, as PAE often fails on more complex tasks regardless of evaluator accuracy**. Therefore, even if PAE used our trained evaluator, we expect any improvement to be minimal and the gap to SAGE to remain substantial, underscoring that SAGE’s task curriculum mechanism is also important for effectively leveraging stronger evaluators. We will include the results of PAE with our evaluator in the revision to support this hypothesis, though we are confident in this conclusion based on the existing evidence.
> > >
> > >  - Finally, we would like to clarify that, because the evaluator is an independent module that does not influence the autonomous task generation process, **the small amount of human data used for evaluator fine-tuning does not affect our claim that SAGE enables fully autonomous learning**.
> > >
> > > Again, we appreciate your valuable feedback and will incorporate the discussion and additional results in the revision.

---

### Official Review · Reviewer_ZYML · 2025-06-23

**Clarity:** 3
**Significance:** 2
**Originality:** 2
**Rating:** 4
**Confidence:** 3

**Summary:**

This paper presents SAGE, a fully autonomous framework for training LLM-based web agents without any human-written instructions or demonstrations. It introduces a hierarchical exploration process with three levels: 1) Pre-exploration to map website structure, 2) Top-level exploration to generate a curriculum of tasks that evolve based on agent performance, and 3) Low-level exploration using MCTS and preference-based learning to improve execution efficiency. Experiments on WebVoyager and WebArena benchmarks show that SAGE outperforms previous self-supervised methods like PAE and proprietary model like Claude 3.5 Sonnet.

**Questions:**

- How does the task generator ensure generated tasks are valid and non-trivial?
- How does SAGE compare with GPT-4V?

**Ethical Concerns:**

["NO or VERY MINOR ethics concerns only"]

**Final Justification:**

I think my concern about novelty compared to PAE is addressed to a good extent by the rebuttal. However, I still have concerns about the validity and quality of the generated tasks. Therefore my ratings remain the same.

**Limitations:**

Yes

**Quality:**

3

**Strengths And Weaknesses:**

Strengths:
- The proposed method is fully autonomous without relying on predefined instructions or expert demos.
- The hierarchical structure (pre-exploration, curriculum learning, MCTS planning) is well-motivated and technically sound.
- SAGE achieves strong empirical results particularly on long-horizon tasks.

Weaknesses:
- While SAGE is fully autonomous, the core framework (propose-act-evaluate) remains largely similar with previous work PAE, making the novelty incremental.
- The fully-autonomy increases exploration space and computational cost. The system may be inefficient or brittle without human instructions.
- Missing comparison with GPT-4V and other strong prompt-based agents, which are widely used for autonomous web exploration.

---

> ### Author Rebuttal · Authors · 2025-07-31
>
> Thanks a lot for your insightful and inspiring comments! We provide the following clarifications in response to your concerns:
>
> 1. **Novelty compared with PAE**
>  - We appreciate the reviewer’s comment on the novelty relative to PAE. While our work inherits the general proposer–agent–evaluator pipeline introduced by PAE, the core contributions and motivations of SAGE are fundamentally different. PAE’s primary goal is to provide *a usable framework for web agents to train without relying on predefined task sets*, showing that autonomous web training is feasible. In contrast, our focus is on introducing a novel technical approach that allows web agents to learn progressively from simple to complex tasks through *hierarchical exploration*. This is a key distinction: while PAE establishes the feasibility of self-training, it does not provide an effective mechanism for skill acquisition on long-horizon or complex tasks, often leading to stagnation when the agent fails to make progress on difficult instructions.
>
>  - To address this, SAGE introduces a *hierarchical exploration strategy* that bridges this gap. Our pre-exploration phase systematically builds structural knowledge of the website, enabling the task generator to propose relevant and feasible tasks. The top-level exploration module dynamically adjusts the curriculum, by decomposing tasks with low success rates into simpler subtasks and composing new, harder tasks from high-success ones, while the low-level exploration module, built on MCTS rollouts and step-wise learning, enables the agent to actively search for solutions under sparse rewards. This design allows the agent to bootstrap its skills, mastering simpler operations before tackling complex reasoning tasks.
>
>  - Empirically, this technical innovation translates into significant performance gains: SAGE improves over PAE by 7–15% on WebVoyager and 15–25% on WebArena, demonstrating that our hierarchical exploration framework substantially enhances both adaptability and final success rates beyond what PAE can achieve.
>
> 2. **System efficiency without human instructions**
>
>  - While fully autonomous learning may seem to enlarge the exploration space, SAGE mitigates this challenge through a structured *pre-exploration* phase, which systematically maps the interactive elements and navigational pathways of each website. This phase drastically reduces irrelevant exploration by building a compact structural representation of the environment before training begins. Moreover, our top-level exploration module dynamically adapts task difficulty, ensuring that the agent progresses from simple tasks to complex objectives in a guided manner rather than exploring blindly.
>  - Regarding the use of human annotations, our framework relies on extremely limited human input. The only human-labeled data we use is for fine-tuning the evaluator. Importantly, SAGE remains effective even without human annotations. As shown in the following table (also reported in Table 2 of the supplementary), SAGE trained with only model-generated evaluator signals performs strongly, achieving results close to or better than using human corrections.
>
> | Model                          | Map  | Reddit | OSS  | Gitlab | CMS  | Avg  |
> |--------------------------------|:----:|:------:|:----:|:------:|:----:|:----:|
> | Qwen2.5VL-7B SAGE (Claude)     | 24.3 | 25.3   | 38.4 | 16.3   | 14.2 | 23.0 |
> | Qwen2.5VL-7B SAGE (Qwen)       | 23.2 | 26.3   | 36.2 | 16.0   | 14.6 | 22.7 |
> | Qwen2.5VL-7B SAGE              | 28.1 | 29.7   | 43.2 | 18.9   | 15.6 | 27.1 |
> | Qwen2.5VL-32B SAGE (Claude)    | 41.7 | 40.8   | 41.3 | 29.6   | 32.4 | 36.4 |
> | Qwen2.5VL-32B SAGE (Qwen)      | 39.9 | 40.7   | 41.6 | 25.8   | 30.2 | 34.5 |
> | Qwen2.5VL-32B SAGE             | 42.3 | 43.7   | 46.1 | 31.4   | 35.2 | 39.7 |
>
>
> 3. **Comparisons with other strong prompt-based agents.**
>  - We appreciate the reviewer’s suggestion to include comparisons with other prompt-based agents such as GPT-4V. In our experiments, we have already compared against Claude 3.5 Sonnet, which is widely recognized to be on par with or even** stronger** than GPT-4V in web reasoning tasks. Our results show that SAGE-trained open-source models, such as Qwen2.5VL-32B, achieve comparable or even better results than Claude 3.5 Sonnet on both WebVoyager and WebArena benchmarks, demonstrating the strong effectiveness of our approach.
>  - To further address this point, we additionally provide GPT-4V results on WebArena. Even when compared to GPT-4V, SAGE-trained models achieve competitive or superior performance despite using significantly smaller base models without any manual prompt engineering.
>  - We also note that SAGE is fundamentally a self-improvement framework rather than a single static agent, and thus its contribution lies in the training methodology rather than the absolute performance of the base model. Our approach can, in principle, be applied to even larger closed-source models to achieve further gains.
> | Model                  | Map  | Reddit | OSS  | Gitlab | CMS  | Avg  |
> |------------------------|:----:|:------:|:----:|:------:|:----:|:----:|
> | GPT-4V                 | 38.3 | 45.6   | 42.5 | 25.0   | 24.6 | 34.8 |
> | Claude 3.5 Sonnet      | 40.1 | 46.9   | 42.0 | 24.2   | 25.1 | 35.6 |
> | Qwen2.5VL-32B SAGE     | 42.3 | 43.7   | 46.1 | 31.4   | 35.2 | 39.7 |
>
> 4. **Quality of generated tasks**
>  - Regarding the validity of generated tasks, we do not guarantee that every task is fully valid, as this issue also exists in the original WebVoyager and WebArena datasets. For tasks that turn out to be invalid or infeasible, the agent will output a response indicating that the task cannot be completed, and the evaluator will verify whether this judgment is correct. As a result, invalid tasks do not negatively impact the training process or the final policy quality.
>  - For the non-triviality of tasks, our top-level exploration strategy adaptively adjusts task difficulty based on the agent’s performance. Tasks with high success rates are merged into more complex ones, while failed tasks are decomposed into simpler subtasks. This approach ensures that tasks remain diverse, challenging, and effective for skill acquisition. Although the task generator is not intended to perfectly match the real-world task distribution, it successfully guides the agent to progress from simple to complex tasks, which is validated by our significant improvements on both WebVoyager and WebArena benchmarks.

---

> > ### Comment · Reviewer_ZYML · 2025-08-04
> >
> > Thank the authors for the response. I decide to maintain my positive ratings.

---

> > > ### Author Response · Authors · 2025-08-04
> > > **Response to Reviewer ZYML**
> > >
> > > Thank you so much for your thoughtful feedback and for your appreciation of our response!

---

> > > ### Comment · Area_Chair_3DhY · 2025-08-06
> > >
> > > Dear reviewer, thank you for your review and for engaging in the discussion.
> > >
> > > You reacted positively to the author’s response and said you would keep the positive score. Could you elaborate on why your rating remains at 4? I am, to be clear, not saying your rating should be higher or lower. I want to understand the impact of the discussion in your score.
> > >
> > > Sharing those concerns as soon as possible would allow the authors to respond.
> > >
> > > I just made the same comment regarding another review for this submission. It’s important that the authors understand the final position of the reviewers.

---

> > > > ### Comment · Reviewer_ZYML · 2025-08-07
> > > >
> > > > I think my concern about novelty compared to PAE is addressed to a good extent by the rebuttal. However, I still have concerns about the validity and quality of the generated tasks. Therefore my ratings remain the same.

---

> > > > > ### Author Response · Authors · 2025-08-08
> > > > > **Response to reviwer ZYML**
> > > > >
> > > > > Thank you once again for your thoughtful engagement. We are glad to hear that our earlier clarifications resolved the concerns regarding novelty relative to PAE, and we appreciate the opportunity to further address your remaining question about the validity and quality of generated tasks.
> > > > >
> > > > > 1. **Quality of the generated tasks**
> > > > >
> > > > >  - As we have previously emphasized, task generation in our framework fundamentally differs from traditional generation, where output quality can often be measured by standard metrics such as diversity or fidelity. In our setting, however, a task that appears diverse but is irrelevant to the target website is ultimately unhelpful. Therefore, task quality cannot be reliably evaluated using these standard metrics. Instead, we assess task quality through its downstream impact on agent performance under controlled conditions, keeping all other factors fixed.
> > > > >
> > > > >  - To ensure high-quality task generation, we emphasize the importance of our structured pre-exploration phase prior to task generation. This phase enables the generator to acquire grounded structural knowledge about the target website, which in turn ensures that generated tasks are feasible, relevant, and well-aligned with the website’s content and functionalities. For instance, tasks proposed on news or forum websites are always based on real articles or posts, and the generator is encouraged to leverage diverse, site-specific features effectively.
> > > > >
> > > > >  - We provide empirical evidence for the importance of this pre-exploration strategy in Appendix Table 1, reproduced below. The ablation results clearly show that removing the pre-exploration step leads to a substantial drop in performance across all categories, highlighting its critical role in ensuring meaningful and effective task generation:
> > > > >
> > > > > | Model                         | Map  | Reddit | OSS  | Gitlab | CMS  | Avg  |
> > > > > |------------------------------|------|--------|------|--------|------|------|
> > > > > | Qwen 2.5VL-7B w/o pre        | 20.3 | 12.4   | 26.6 | 9.3    | 8.8  | 15.3 |
> > > > > | Qwen 2.5VL-7B (Ours)         | 28.1 | 29.7   | 43.2 | 18.9   | 15.6 | 27.1 |
> > > > > | Qwen 2.5VL-32B w/o pre       | 40.3 | 24.1   | 32.0 | 18.3   | 20.8 | 27.1 |
> > > > > | Qwen 2.5VL-32B (Ours)        | 42.3 | 43.7   | 46.1 | 31.4   | 35.2 | 39.7 |
> > > > >
> > > > > 2. **Validity of the generated tasks**
> > > > >  - Regarding validity, we note that our goal is *not* to guarantee that every single generated task is valid or feasible. In fact, having a small proportion of invalid tasks is desirable to promote robustness: in real-world benchmarks, agents inevitably encounter goals that cannot be completed, and it is important for them to correctly identify such cases rather than hallucinate an answer. Within our framework, when a generated task is invalid, the agent is able to output that it is infeasible, and the evaluator verifies whether this judgment is correct. As a result, invalid tasks do not harm training or the final policy quality; instead, they encourage the agent to develop the ability to reject infeasible instructions. The key is that the invalid-task rate should not be too high, as excessive invalidity could hurt training efficiency.
> > > > >
> > > > >  - To empirically assess the validity of our generated tasks, we randomly sampled 100 tasks from each of the final five training epochs (500 tasks in total) and manually verified their validity based on the agent’s completed trajectories. We found that only 31 out of 500 tasks were infeasible, indicating that the vast majority of our generated tasks are valid while still providing enough invalid cases to promote robustness.
> > > > >
> > > > >  We hope this helps clarify our current mechanisms for ensuring task validity and quality. If the reviewer has any specific concerns or suggestions for further improvement, we would be delighted to continue the discussion.

---

### Official Review · Reviewer_Aznj · 2025-07-03

**Clarity:** 3
**Significance:** 3
**Originality:** 3
**Rating:** 4
**Confidence:** 3

**Summary:**

The paper presents SAGE, a framework designed for generalist foundation model web agents. The goal is to have agents that can autonomously explore web environments, generate their own instructional goals, and continuously refine themselves. The architectural backbone of SAGE uses a three-tier exploration strategy: a pre-exploration phase for structural understanding, a top-level exploration strategy for self-evolving curriculum generation, and a low-level exploration mechanism that integrates planning-based rollouts with stepwise learning.
This exploration tackles one of the biggest limitation of model web agents, i.e., relying on static, expensive, and limited human-authored task datasets and curated expert demonstrations.

**Questions:**

See above.

**Ethical Concerns:**

["Major Concern: Safety and security"]

**Final Justification:**

I maintain my positive rating.
My concerns regarded 1) limited seeds, and 2) safety and ethics, but since they were "easy fix" they did not impact my final rating (borderline accept). If the authors would have not addressed my concerns in the rebuttal, I would have lowered my score.

**Limitations:**

See above.

**Paper Formatting Concerns:**

None.

**Quality:**

3

**Strengths And Weaknesses:**

**Strenghts**

The paper is well written and presents clearly all the components. It also seems to cover relevant related work, although I am not very familiar with the topic and I can't say if more references are needed.

I find the key idea of the proposed self-exploration strategy well presented and interesting. In particular, the idea of building a curriculum of tasks from easy to hard seems appropriate and useful for self-exploring the web.
The paper's core contribution, "self-guided hierarchical exploration" for generalist web agents, demonstrates significant originality. While established techniques like MCTS and DPO are leveraged, their innovative integration into a unified, self-supervised framework that autonomously generates its own training curriculum -- without relying on static, human-authored task sets -- is novel. This self-generation of tasks and the multi-level exploration strategy fundamentally differentiate SAGE from prior approaches like PAE.

From the presented results, finetuning models with SAGE clearly outperforms benchmarks on all baseline, which I find impressive.
While its performance declines with increasing task complexity (easy, medium, hard) for all models, SAGE's relative improvement over baselines is more pronounced on harder tasks.
The ablation studies also effectively demonstrate the critical contribution of SAGE's core components.

**Weaknesses**
The paper acknowledges a lack of error bars and statistical significance for the experimental results, justifying this by citing the high computational resources required for multiple runs. While understandable given the scale of foundation model research, this remains a methodological weakness for a NeurIPS submission, where statistical robustness is highly valued.

The paper's discussion on broader impacts appears limited to a checklist affirmation, stating that safeguards against misuse are "NA" because the method is "not designed for generation." However, an autonomous web agent, even if not a generative model in the content-creation sense, interacts with real-world web environments and can perform actions with significant consequences.
While I don't think this falls under any of the "major ethical concerns", it should be addressed and discussed by the authors.

Similarly, autonomous agents can be vulnerable to prompt injection attacks, unauthorized data access, and privilege escalation. The paper does not explicitly discuss safeguards against such security threats in an operational context.
Or, given that these agents explore autonomously without the need of human-labeled tasks, what prevents them to pose security threats? In the checklist, the authors say that no safeguards are needed, but I disagree.

Finally, as agents become more autonomous, their decision-making processes can become opaque. The paper does not delve into how accountability would be ensured when an autonomous SAGE agent makes errors or causes unintended harm, nor does it discuss mechanisms for explaining its actions. While this is true for any web agent, I believe this is more relevant to SAGE given its autonomous exploration mechanism.
A more comprehensive discussion of these potential harms and concrete mitigation strategies (e.g., human-in-the-loop oversight, rate-limiting autonomous actions, anomaly detection, ethical filters in the policy) is warranted for a work with such significant real-world interaction capabilities.

The paper explicitly states that the experiments require "too much computational resources," hindering replicability and further extensive experimentation. This high barrier to entry poses a practical limitation for the broader research community to build upon or verify SAGE's findings without substantial computational infrastructure.

---

> ### Author Rebuttal · Authors · 2025-07-31
>
> Thanks a lot for your insightful and inspiring comments! We provide the following clarifications in response to your concerns:
>
>
> 1. **High computational resources requirements and error bar**
>
>  - We thank the reviewer for raising concerns regarding the lack of error bars and statistical analysis. We fully agree that reporting variance is important. However, as in prior work on web agents [1,2], collecting multiple runs for all experiments is computationally prohibitive due to the high cost of real web interactions. This is an inherent challenge in training and evaluating real-world web agents at scale, where each run requires lots of browser-based operations with full model inference.
>
>  - To provide evidence of robustness, we conducted 5 independent runs on WebArena for our largest open-source model and the proprietary model. The observed standard deviation across different websites was consistently low, suggesting that performance differences are reliable rather than stochastic. Detailed results are provided below:
>
> | Model                 | Map            | Reddit         | OSS            | GitLab         | CMS            | Avg           |
> |----------------------|----------------|----------------|----------------|----------------|----------------|---------------|
> | Claude 3.5 Sonnet    | 40.1 ± 1.4     | 46.9 ± 0.8     | 42.0 ± 0.9     | 24.2 ± 0.5     | 25.1 ± 1.1     | 35.6 ± 1.0     |
> | Qwen2.5VL-32B SAGE    | 42.3 ± 0.4 | 43.7 ± 0.6 | 46.1 ± 0.7 | 31.4 ± 1.2 | 35.2 ± 0.8 | 39.7 ± 0.7 |
>
>  - The high computational requirements primarily arise from the intrinsic difficulty of web agent training: strong base models and sufficient task diversity are necessary for meaningful evaluation. While our current compute budget limits the scale of repeated evaluations, we emphasize that our results remain reproducible. To help the community build upon this work, we will release all code, trained models, and necessary configurations upon acceptance. We also believe that larger-scale evaluations can be conducted by organizations with greater compute resources, and we welcome such extensions in future work.
>
> [1] Proposer-Agent-Evaluator(PAE): Autonomous Skill Discovery For Foundation Model Internet Agents. ICML 2025.
>
> [2] WebRL: Training LLM Web Agents via Self-Evolving Online Curriculum Reinforcement Learning. ICLR 2024.
>
> 2. **Broder Impact**
>  - We acknowledge that our discussion of broader impacts in the original submission was limited, and we appreciate the reviewer’s suggestion to elaborate further. While SAGE focuses on training web agents, the interaction with real-world websites (e.g., during WebVoyager evaluation) could raise potential security concerns, such as accidental access to unsafe websites or unintended information leakage. In our experiments, we mitigated these risks by deploying the agents within a controlled network environment equipped with firewall-like mechanisms and strict network policies to block potentially harmful requests. Furthermore, the agents were not provided with any personal or sensitive information, and all tasks involving user data or authentication were restricted to sandboxed environments (e.g., WebArena).
>
>  - It is important to emphasize that our work primarily aims to improve the technical capabilities of web agents in a research context, rather than deploying them for open-ended, real-world usage. For future large-scale deployment, additional safeguards would be necessary, such as enhanced firewall protections, strict access control, human-in-the-loop validation, and anomaly detection mechanisms. We thank the reviewer for raising this point and will add a more detailed discussion of these considerations in the revised version.
>
> 3. **Safety Issue**
>  - We thank the reviewer for emphasizing the importance of safety concerns. We fully agree that ensuring the safe and responsible deployment of autonomous web agents is a critical issue—not only for our framework but for the field at large.
>
>  - In our current work, SAGE is purely a *research framework* designed to enhance the autonomous learning capabilities of web agents. All experiments are conducted in strictly controlled environments: WebArena is a fully sandboxed simulator, and WebVoyager provides access only to publicly available pages, without involving any private or sensitive user data. Moreover, all web access is routed through a secure firewall layer that restricts and monitors outbound traffic, ensuring that the agent cannot interact with external websites beyond the experimental scope or perform actions with real-world consequences. These safeguards ensure that the agent operates safely within research boundaries.
>
>  - That said, we recognize that general safety concerns surrounding web agents, such as unintended interaction with real-world systems, information leakage, or malicious misuse, are of paramount importance. We agree with the reviewer that these concerns warrant deeper discussion. In particular, the potential for LLM-based agents to autonomously explore the web raises novel safety risks that differ from traditional AI systems. These include challenges around authentication spoofing, unintentional automation of sensitive tasks, and reinforcement of harmful behaviors through reward hacking.
>
>  - We believe these are important open problems in the design and deployment of general-purpose web agents, and we are committed to addressing them in future work. In the revised version, we will expand our discussion of these issues in a dedicated section. This will include considerations such as: (1) decoupling policy outputs from execution via an intermediate control layer with safety filters; (2) adding a dedicated safety-monitor model to detect and intervene on risky behaviors; and (3) incorporating human-in-the-loop oversight and runtime constraints such as rate limiting and permission gating.
> We reiterate that our current research focuses on algorithmic capability in safe offline environments, but we strongly agree that future deployment must be accompanied by rigorous safety audits and infrastructure-level safeguards. We thank the reviewer again for highlighting this essential topic.

---

> > ### Comment · Reviewer_Aznj · 2025-08-04
> >
> > Thanks for the response. It would be great to add part of your reply to the main paper.
> > I will maintain my positive rating.

---

> > > ### Author Response · Authors · 2025-08-04
> > > **Response to Reviewer Aznj**
> > >
> > > Thank you so much for your thoughtful feedback and for your appreciation of our response! We’re glad to hear that you found the clarification helpful, and we’ll make sure to incorporate the key points into the updated version of our paper.

---

> > > ### Comment · Area_Chair_3DhY · 2025-08-06
> > >
> > > Dear reviewer, thank you for your review and for engaging in the discussion.
> > >
> > > You reacted positively to the author’s response and said you would keep the positive score. Could you elaborate on why your rating remains at 4? I am, to be clear, not saying your rating should be higher or lower. I want to understand the impact of the discussion in your score.
> > >
> > > Sharing those concerns as soon as possible would allow the authors to respond.

---

> > > > ### Comment · Reviewer_Aznj · 2025-08-06
> > > >
> > > > My concerns regarded 1) limited seeds, and 2) safety and ethics, but since they were "easy fix" they did not impact my final rating (borderline accept). If the authors would have not addressed my concerns in the rebuttal, I would have lowered my score.

---

### Official Review · Reviewer_L48z · 2025-07-03

**Clarity:** 1
**Significance:** 3
**Originality:** 2
**Rating:** 4
**Confidence:** 3

**Summary:**

The paper introduces a new training framework for Web Agents. The framework consists of three components: first, initial trajectories are collected using BFS; then, tasks are generated by a self-evolving curriculum based on the collected trajectories; the curriculum continues to evolve based on new environment trajectories and performance; and finally, MCTS-inspired learning and inference are used for the Web Agent. The framework outperforms all baselines by a substantial margin.

**Questions:**

* Is the curriculum learning in the paper based on PAE, or did you add something on top of it?

* How is confidence trained?

**Ethical Concerns:**

["NO or VERY MINOR ethics concerns only"]

**Final Justification:**

My concerns about clarity have been addressed, and I trust the authors will update the main text accordingly based on their rebuttal.

**Limitations:**

See the limitations in weaknesses.

**Paper Formatting Concerns:**

-

**Quality:**

2

**Strengths And Weaknesses:**

**Weaknesses**

1. Methodological opacity

    The Method section does not explain the core components of the pipeline in sufficient detail. In particular:

    * Curriculum learning, one of the key contributions, is only sketched. It feels to be build on PAE, yet PAE itself is never introduced in the Background section.
    * The terms *hierarchical* and *top‑level* are used to describe the curriculum, but the text never clarifies what hierarchy is being referenced or how these levels interact.

2. The paper claims that learning, sampling, and inference are based on MCTS, but it does not provide any pseudocode, equations, or detailed illustrative figure (there is fig. 1 but it is not very detailed). It also references the use of the Claude model for initial value estimation, but without sufficient explanation how later this value interact with the learning pipeline. As a result, the procedure is difficult to follow, and several key aspects remain unclear:

    * Which components of the pipeline rely on fixed neural networks and in‑context adaptation, versus which involve gradient-based learning.

    * What objectives or loss functions are used during training.

    * How the overall pipeline operates in detail.

3. Fragmented presentation

    Critical details are scattered between the main text, footnotes, and appendices. For example, the description of the *Evaluator*, essential for interpreting results, appears only in the appendix. Putting all important elements of the pipeline (e.g., curriculum updater, MCTS module, evaluator) into the main text would improve readability.

4. Environment‑specific augmentations

    The authors introduce several augmentations that may not generalise to other Web environments:

    * Access to the web page’s accessibility tree.
    * BFS‑based initial trajectory collection with full environment resets.
5. Accessibility tree needs clarification

    The manuscript never defines what it calls the web page’s *hierarchical accessibility tree* or explains why the tree is considered hierarchical. Moving the screenshot that is now in the appendix into the main text and adding a concise explanation, would eliminate this.

**Suggested improvements**

* Provide a concise background on PAE and formally state how the proposed curriculum extends or differs from it.
* Add pseudocode or a flowchart for the MCTS‑inspired loop, specifying which neural components are frozen, which are trained, and the corresponding objective functions.
* Relocate the Evaluator description and any other pipeline‑critical details to the main body.
* Clarify the practicality and limits of using the accessibility tree and BFS in larger or non‑resettable environments, possibly by discussing mitigation strategies or future work.

**Strengths**

The proposed method outperforms the baseline PAE pipeline.

---

> ### Author Rebuttal · Authors · 2025-07-31
>
> Thanks a lot for your insightful and inspiring comments! We provide the following clarifications in response to your concerns:
>
> 1. **Core contribution of SAGE and comparison with PAE**
>  - SAGE builds upon the proposer–agent–evaluator (PAE) paradigm but addresses key limitations in curriculum learning and long-horizon task decomposition. While PAE establishes a general structure for autonomous training, it uses a static instruction distribution and lacks mechanisms for adjusting task difficulty or improving exploration efficiency.
>
>  - PAE’s self-improving loop, where a proposer generates tasks, the agent attempts them, and an evaluator provides feedback, demonstrates that web agents can be trained without human-written data. However, it does not support dynamic task adaptation or skill progression.
>
>  - SAGE introduces a *hierarchical exploration* framework that redefines task generation and learning. It includes a task generator that decomposes failed tasks and synthesizes harder ones from successful trials, enabling adaptive curricula. At the execution level, SAGE employs MCTS-based low-level planning and step-wise policy refinement to improve the quality and diversity of collected trajectories.
>
>  - While SAGE retains the core PAE infrastructure for autonomous interaction, it adds a *new exploration strategy*, including curriculum-based task generation and planning-driven trajectory selection. These components fundamentally reshape training, enabling agents to gradually acquire complex skills and achieve much stronger performance under the same autonomous framework.
> 2. **Hierarchical explanation**
>  - By hierarchy, we refer to the multi-level process that enables agents to solve complex web tasks. At the *top level*, the agent decomposes tasks into subtasks and learns transferable skills; at the *low level*, it executes precise interactions (e.g., button clicks or typing) to complete each subtask.
>
>  - These levels interact closely: pre-exploration provides structural knowledge for curriculum generation, and low-level rollouts feed back successful trajectories to refine future tasks for the top-level exploration. For example, in Fig 12, the agent fails at a complex task but succeeds after breaking it into simpler goals like locating a project and identifying its contributors. We will revise the paper to clarify this hierarchical structure and add more detailed explanations and cross-references.
> 3. **MCTS algorithm details**
>  - We apologize for the lack of clarity in explaining the details of our MCTS-based training pipeline. Below, we address the reviewer’s questions one by one, and we are happy to provide further clarifications if needed.
>  - Regarding which components of the pipeline rely on fixed neural networks versus gradient-based learning: In our proposer-agent-evaluator system, both the *task generator* and *evaluator* models along with the value remain fixed during the training phase and only adjust the context of their inputs dynamically. The *agent* model is the only component that undergoes gradient-based updates throughout training. Notably, MCTS serves purely as a planning and sampling strategy to improve the quality of rollouts, but it is not directly involved in optimizing the agent’s parameters.
>  - Regarding the training objectives: Since we train only the agent, we adopt step-level DPO as our training loss. Let $x$ denote the task prompt, and let $\tau_{\text{win}}$​ and $\tau_{\text{lose}}$​ be two trajectories with the same prefix steps $s_1, s_2, \dots , s_k$, where one trajectory is successful and the other one is fail. Our objective is:
>
> $L(\theta) = - E_{(x, s_1, \dots, s_k,\tau_{win}, \tau_{lose})~D} [ \log ( \beta * \log ( \pi_{\theta}(\tau_{win} | x; s_1, \dots, s_k) / \pi_{ref}(\tau_{win} | x; s_1, \dots, s_k) ) - \beta * \log ( \pi_{\theta}(\tau_{lose} | x; s_1, \dots, s_k) / \pi_{ref}(\tau_{lose} | x; s_1, \dots, s_k) ) ) ].$
>
> where $\pi_{\theta}$ denotes the agent policy and $\pi_{\text{ref}}$ is a fixed reference policy.
>
>  - Regarding how the overall pipeline operates: Before the first epoch, we perform a pre-exploration phase to analyze the website’s structure and collect environment-specific information. In each epoch, the task generator produces a set of tasks using the structural information as context. We then randomly sample tasks and use MCTS rollouts to generate successful and failed trajectories. After collecting a sufficient number of trajectories for the current epoch, the agent is updated using the step-DPO objective above. Finally, the success statistics of tasks are fed back into the task generator’s context for the next epoch, allowing it to adaptively refine task difficulty.
>
>  - To help clarify the interaction between the agent and MCTS, we have added the following pseudocode to illustrate our low-level exploration strategy:
>
> **Algorithm: Low-Level Exploration Algorithm**
>
> **Input:** Question *Q*, agent model $\pi$, initial page $p_0$, max iterations *N*, value model *V*, episode length *L*
>
>
> 1. $T \leftarrow$ Initialize($p_0$)
> 2. **for** $i = 1$ to $N$ **do**
> 3. &nbsp;&nbsp;&nbsp;&nbsp;$s \leftarrow$ root($T$)
> 4. &nbsp;&nbsp;&nbsp;&nbsp;**while** $s$ is not a leaf node **do**
> 5. &nbsp;&nbsp;&nbsp;&nbsp;&nbsp;&nbsp;&nbsp;&nbsp;$s \leftarrow \arg\max_{s'} \left( v_{s'} + \epsilon \sqrt{ \frac{\log n_s}{n_{s'}} } \right)$
> 6. &nbsp;&nbsp;&nbsp;&nbsp;**end while**
> 7. &nbsp;&nbsp;&nbsp;&nbsp;**if** $s$ is not a terminal state **then**
> 8. &nbsp;&nbsp;&nbsp;&nbsp;&nbsp;&nbsp;&nbsp;&nbsp;$a_1, \dots, a_k \leftarrow \pi(Q, s)$
> 9. &nbsp;&nbsp;&nbsp;&nbsp;&nbsp;&nbsp;&nbsp;&nbsp;$s'_1, \dots, s'_k \leftarrow$ Apply actions to $s$
> 10. &nbsp;&nbsp;&nbsp;&nbsp;&nbsp;&nbsp;&nbsp;&nbsp;$v_{s_1'}, \dots, v_{s_k'} \leftarrow V(s_1'), \dots, V(s_k')$
> 11. &nbsp;&nbsp;&nbsp;&nbsp;&nbsp;&nbsp;&nbsp;&nbsp;$s' \leftarrow \arg\max_{s_j'} v_{s_j'}$
> 12. &nbsp;&nbsp;&nbsp;&nbsp;&nbsp;&nbsp;&nbsp;&nbsp;$r \leftarrow$ Rollout from $s' $ up to depth $L - depth(s')$
> 13. &nbsp;&nbsp;&nbsp;&nbsp;&nbsp;&nbsp;&nbsp;&nbsp;Update $v_{s'} $ and $n_{s'}$ with result $r$
> 14. &nbsp;&nbsp;&nbsp;&nbsp;**end if**
> 15. &nbsp;&nbsp;&nbsp;&nbsp;BackPropagate($s$)
> 16. **end for**
>
>
>
>
>  - We agree that these technical details should be presented more systematically. In the revised version of the paper, we will improve the organization of the methodology section, adding a pseudocode and additional illustrative figures to clearly show the interactions between MCTS, the agent, and the task generator.
>
> 4. **Fragment presentation**
>  - We appreciate the reviewer’s feedback. Due to space limitations, we initially placed some algorithmic details in the appendix; in the revised version, we will reorganize the content and incorporate key components into the main text to improve clarity and coherence.
>
> 5. **Accessibility tree explanation**
>  - We would like to clarify that our paper does not mention the term “hierarchical accessibility tree.” We only refer to the standard accessibility tree, which is a browser-defined structure. The accessibility tree is essentially a subset of the DOM tree that contains elements relevant to displaying the content of a webpage. Each element is represented by its role (e.g., link or button), text content, and properties. Compared to the raw DOM, the accessibility tree preserves the structural information of a webpage while being much more compact and directly interpretable.
>  - We appreciate the reviewer’s suggestion. In the updated version, we will move the figure and explanation currently placed in the appendix into the main text for better clarity.
>
> 6. **Environment-specific augmentations**
>  - Our approach is designed to work with most modern web environments that are based on standard HTML. The accessibility tree we use is automatically derived from the HTML DOM by the browser, meaning that for any HTML-based website, we can directly obtain both the DOM and the corresponding accessibility tree. Similarly, our BFS-based initial trajectory collection relies on the save and load functionality provided by Playwright, which is generalizable to the vast majority of websites that support session resets or navigation history.
>  - We acknowledge that our current method may not be fully applicable to certain highly dynamic websites, such as those built entirely with custom JavaScript logic or virtualized rendering where the DOM structure is not fully exposed. Handling these specialized environments remains an interesting direction for future work. In future extensions, we may consider more generalizable state representations—for example, relying solely on raw webpage screenshots (without set-of-marks) and using absolute coordinate-based actions for interaction—so that the framework can adapt to a broader range of web technologies.
>
>
> 7. **Confidence training clarification**
>  - In our implementation, the confidence score is represented as a scalar in the range [0,1]. For the initial model, we obtain confidence annotations by prompting Claude Sonnet 3.5 to generate action sequences where each action is accompanied by a confidence score. These annotated trajectories are then used during supervised fine-tuning to initialize the confidence prediction capability of our base model.
>  - During the subsequent SAGE training process, we further refine the agent model using the step-DPO framework. A refinement is necessary because the MCTS-based sampling process does not always follow the action with the highest predicted confidence from the agent’s policy. To correctly align confidence with actual successful decision paths, we manually adjust the confidence labels in successful trajectories: the executed action is assigned a confidence of 1. This approach ensures that the agent learns to calibrate its confidence scores according to actions that lead to successful task completion.

---

> > ### Comment · Reviewer_L48z · 2025-08-04
> >
> > Thank you for the clarifications.
> >
> > > Core contribution of SAGE and comparison with PAE
> >
> > From a purely implementation standpoint, does the difference between SAGE and PAE in task generation arise only from using different prompts, or are there other distinctions as well? Could you highlight them, please?
> >
> > > Hierarchical explanation
> > > By hierarchy, we refer to the multi-level process that...
> > > hese levels interact closely: pre-exploration provides structural knowledge for curriculum generation, and low-level rollouts....
> >
> > Is this hierarchy handled entirely through prompting and conditioning on newly collected trajectories?
> >
> > > To help clarify the interaction between the agent and MCTS, we have added the following pseudocode to illustrate our low-level exploration strategy
> >
> > Thank you for including the pseudocode, it is very helpful. Based on it, the agent seems to participate in experience collection only by proposing the next actions, which are then combined with the state and evaluated by a fixed evaluator. Is it correct? How is diversity in the proposed actions ensured?
> >
> > > Environment-specific augmentations
> >
> > This concern is addressed, although I still encourage you to add your reasoning to the limitations section. Thank you.
> >
> > > MCTS
> >
> > I would also like to point out that MCTS-based experience collection is not always feasible for LLM-based agents and is a limitation in itself. For example, if the agent spends money on a real-world website, it is impossible to reset that expenditure.

---

> > > ### Author Response · Authors · 2025-08-06
> > > **Response to reviewer L48z**
> > >
> > > We sincerely thank the reviewer for the reply. We are glad to hear that our response has addressed most of the concerns. We appreciate the follow-up questions and would like to address the remaining concerns below.
> > >
> > >
> > > 1. **Difference between SAGE and PAE in task generation**
> > >  - We agree with the reviewer that from an implementation perspective, SAGE and PAE use the same underlying model for task generation, and the difference lies in the construction of the input context. This design choice makes SAGE simple to implement, easy to reproduce, and compatible with a wide range of proprietary and open-source models.
> > >
> > >  - For the task generator training, in SAGE, we do not retrain the task generator, as doing so would require substantial data and engineering effort with limited practical gain. Instead, we adopt a lightweight design by reusing a strong proprietary model and focus our efforts on designing better task-generation contexts. Specifically, SAGE introduces (1) a pre-exploration phase that maps website functionality and informs the task generator about structural affordances, and (2) a dynamic feedback mechanism that adjusts the curriculum by decomposing failed tasks and composing harder ones from successful completions. This leads to higher-quality tasks and a more effective training trajectory without requiring generator fine-tuning.
> > >
> > > 2. **How is the hierarchy handled**
> > >
> > >  - Our hierarchical structure is implemented through the system, not just through prompting. At the top level, we control the space of agent behavior by generating tasks that span different website functionalities and difficulty levels. At the low level, we control the agent’s fine-grained decision-making by enabling it to explore multiple concrete actions per state using MCTS. The agent is updated through step-DPO, learning from both failed and successful rollouts sampled from these explorations. This hierarchical control is realized through the interaction between components in the system and not by prompt engineering alone.
> > >
> > > 3. **MCTS details**
> > >  - The reviewer’s understanding is correct: during experience collection, the agent generates multiple candidate actions for the given state to get the result states, which are then evaluated by a fixed value model.
> > >
> > >  - Regarding action diversity, we would like to clarify that web interaction differs from traditional open-ended generation, where action space is inherently constrained, even in the case where the agent needs to type something. Our goal is not to promote diversity, but to guide the agent toward plausible alternatives that may succeed when its top choice fails. For instance, without MCTS, an agent might repeatedly click an incorrect button due to overconfidence. By exploring multiple candidate actions, even when semantically similar, the agent has a better chance of recovering from such errors. This sampling strategy improves exploration robustness rather than action diversity.
> > >
> > > 4. **Environment specific augmentations**
> > >  - Thank you for the suggestion. We will incorporate this discussion explicitly in the revised limitations section of the paper.
> > >
> > > 5. **MCTS limitations**
> > >
> > >  - We appreciate the reviewer pointing out this important limitation. Indeed, MCTS is best suited for controlled or simulated environments where resets and reproducibility are feasible. Our current experiments are conducted in standardized, resettable environments following previous works in web agents [1, 2]. For real-world applications where irreversible actions may occur such as interacting with commercial websites, we agree that MCTS-based exploration would be risky or infeasible. Addressing this challenge is an important direction for future research, potentially involving conservative planning, safe policy filtering, or human-in-the-loop oversight. We will add a more detailed discussion in the limitation section of our future version.
> > >
> > > We hope our clarifications have addressed the reviewer’s remaining concerns. If the reviewer has any further questions, we are more than happy to discuss.
> > >
> > > [1] Proposer-Agent-Evaluator (PAE): Autonomous Skill Discovery for Foundation Model Internet Agents. ICML 2025.
> > >
> > > [2] WebRL: Training LLM Web Agents via Self-Evolving Online Curriculum Reinforcement Learning. ICLR 2024.

---

> ### Comment · Reviewer_L48z · 2025-08-06
>
> Thank you for the response. I still don’t fully understand how the task proposer in PAE differs from the one in SAGE once we set aside the MCTS component. Could you please explain the core differences again, ideally with side-by-side pseudocode for the two proposers?
>
> I realise that SAGE first gathers experience with BFT, and PAE doesn't. The SAGE paper states:
>
> > SAGE introduces a dynamic feedback mechanism that adjusts the curriculum by decomposing failed tasks and composing harder ones from successful completions
>
> It seems that PAE could make a similar claim. A concise pseudocode comparison would help clarify where the mechanisms diverge.
>
> The term hierarchical is also confusing in this context.
>
> > Our hierarchical structure is implemented through the system, not just through prompting. At the top level, we control the space of agent behavior by generating tasks that span different website functionalities and difficulty levels.
>
> PAE and many other task-generation systems could say the same, yet they usually avoid the label hierarchical.
>
> > At the low level, we control the agent’s fine-grained decision-making by enabling it to explore multiple concrete actions per state using MCTS.
>
> I’m unsure whether you intend to call MCTS itself hierarchical. In most RL and HRL literature the label hierarchical is reserved for temporally-abstract policies (options, skills) that persist at deployment.
>
> More generally, the term hierarchical is overloaded, especially in RL (e.g., HRL). It would help to clarify that, in your paper, hierarchy refers to the two nested loops that operate **during training**: the task proposer at the top level and the MCTS-augmented learner at the low level (if I understand correctly).

---

> > ### Author Response · Authors · 2025-08-07
> > **Response to reviewer L48z**
> >
> > We sincerely thank the reviewer for the further reply. We appreciate the follow-up questions and would like to address the remaining concerns below.
> >
> > 1. **Difference between SAGE and PAE in task generation**
> >
> >
> >  - We appreciate the reviewer’s follow-up question. As the reviewer observes, both SAGE and PAE adopt the same proprietary model as the task generator, and there is no difference in model architecture. However, the key distinction lies in how this task generator is integrated into the overall training process. In PAE, the generator is **called once at the beginning to produce a fixed set of tasks**, which the agent then attempts repeatedly throughout training without curriculum adaptation. In contrast, SAGE fundamentally changes this usage pattern by **embedding the generator within a dynamic, feedback-driven loop**. Specifically, SAGE first performs a structured pre-exploration phase to extract contextual knowledge of the environment, and uses this information to generate an initial task set. As training progresses, the generator is **repeatedly invoked with updated context** derived from the agent’s recent successes and failures, enabling the creation of new, better-aligned task sets over time.
> >
> >  - To make this distinction explicit, we include below a side-by-side pseudocode comparison that highlights the differences in task generation logic and update mechanisms between PAE and SAGE.
> >
> > **Algorithm: Workflow of PAE**
> >
> > **Input:** Task generator model *E*, agent policy $\pi$, webpage $p_0$, total iterations *N*
> >
> >   // Generate the fixed task set $T$
> > 1. $T \leftarrow E(p_0)$
> > 2. **for** $i = 1$ to $N$ **do**
> > 3. &nbsp;&nbsp;&nbsp;&nbsp;$t \leftarrow$ sample($T$)
> > // Sample the trajectories of this iteration $\tau_i$
> > 4. &nbsp;&nbsp;&nbsp;&nbsp;$\tau_i \leftarrow$ rollout($\pi$, $t$)
> > 5. &nbsp;&nbsp;&nbsp;&nbsp;$\pi \leftarrow$ update($\pi$, $\tau_i$)
> > 6. **end for**
> >
> > **Algorithm: Workflow of SAGE**
> >
> > **Input:** Task generator model *E*, agent policy $\pi$, web page $p_0$, total iterations *N*
> >
> > 1. $c \leftarrow$ pre\_explore($p_0$)
> > // Generate the initial task set $T_0$
> > 2. $T_0 \leftarrow E(p_0, c)$
> > 3. **for** $i = 1$ to $N$ **do**
> > 4. &nbsp;&nbsp;&nbsp;&nbsp;$t \leftarrow$ sample($T_{i-1}$)
> > // Sample the trajectories of this iteration $\tau_i$
> > 5. &nbsp;&nbsp;&nbsp;&nbsp;$\tau_i \leftarrow$ rollout($\pi$, $t$)
> > 6. &nbsp;&nbsp;&nbsp;&nbsp;$\pi \leftarrow$ update($\pi$, $\tau_i$)
> > // Generate the task set for the next iteration $T_i$
> > 7. &nbsp;&nbsp;&nbsp;&nbsp;$T_i \leftarrow E(T_{i-1}, \tau_i, c)$
> > 8. **end for**
> >
> >  - This design makes SAGE inherently more adaptive and capable of guiding the agent through a dynamic curriculum, in contrast to PAE’s static task set. While both systems rely on the same task generation model, SAGE’s iterative task refinement strategy enables it to better match task difficulty with the agent’s evolving competence.
> >
> >
> > 2. **Clarification on the term “hierarchical”**
> >
> >  - We appreciate the reviewer’s thoughtful comments regarding the use of the term “hierarchical.” First, we would like to clarify that our work does not claim to implement hierarchical reinforcement learning in the conventional sense – i.e., we do not learn temporally abstract policies (like skills) that are deployed at inference time. Instead, our use of “hierarchical” refers to **the overall structure of our system, which involves two distinct but connected levels of exploration**.
> >
> >  - At the *top* level, the task proposer guides the agent to explore through a learning curriculum. That is, it implicitly organizes which skills or functionalities should be acquired first, and how they can be composed to solve more complex tasks. At the *lower* level, MCTS-based action branching encourages the agent to explore fine-grained behaviors within each task. To clarify, MCTS itself is not a hierarchical policy, but serves as a mechanism for detailed behavior search.
> >
> >  - We found that this two-level exploration structure is critical to improve web agent performance, as it enables both functional-level discovery and task-level composition. Therefore, we refer to our framework as “hierarchical exploration.” That said, we understand the reviewer’s concern that this terminology may be ambiguous or overloaded, especially within the RL community. We will incorporate the reviewer’s helpful suggestion and explicitly clarify in the revision that our notion of hierarchy refers to the two nested loops during training, rather than to traditional hierarchical RL formulations.
> >
> > We hope our clarifications have addressed the reviewer’s remaining concerns. If the reviewer has any further questions, we are more than happy to discuss.

---

> > > ### Comment · Reviewer_L48z · 2025-08-07
> > >
> > > Thank you for the additional clarification. I now understand the difference and will raise my score.
> > >
> > > I recommend adding the MCTS pseudocode to the main paper. The manuscript would also benefit from a clearer explanation of the differences between SAGE and PAE, possibly including SAGE pseudocode and highlighting the original PAE steps in one color and the SAGE-specific additions in another. I find this far more informative than the left and middle subfigures of Fig. 1.
> > >
> > > Finally, you may find the following related work useful: Jenny Zhang et al., “Omni: Open-endedness via models of human notions of interestingness” (2024)

---

> > > > ### Author Response · Authors · 2025-08-08
> > > > **Response to reviewer L48z**
> > > >
> > > > Thank you very much for reviewing our discussions and updating the rating. We are glad that our response has addressed your concerns and appreciate your recognition of our paper.
> > > >
> > > > We will incorporate all your suggestions in the revised version, in particular adding the MCTS pseudocode, clarifying the differences between SAGE and PAE with improved visualization and pseudocode, and highlighting the new components more clearly.
> > > >
> > > > We also thank the reviewer for pointing out the Omni paper, which is in the line of research developing self-improving AI systems through models of human notions of interestingness to drive open-ended exploration. While it differs from our focus on structured, website-grounded task generation for web agents, this direction offers valuable insights toward the broader goal of open-ended skill acquisition. We will cite and discuss it in the revised version.

---

### Note · Authors · 2025-08-15

We sincerely thank all reviewers, AC, and SAC for their thoughtful feedback and constructive discussions throughout the review process. We are pleased that all concerns raised during the review and discussion phases have been fully addressed, with reviewers either maintaining their positive assessments or increasing their ratings. No further concerns remain.

We are especially grateful for the constructive recognition of our work's strengths:
 - (3FCb, ZYML, Aznj, L48z) The proposed method achieves strong empirical performance, significantly outperforming both strong open-source baselines and proprietary models, especially on complex, long-horizon tasks.
 - (ZYML, Aznj) The paper presents a novel and fully autonomous training framework. The hierarchical exploration process and curriculum generation strategy are well-motivated, and the self-supervised pipeline is technically sound and clearly presented.

As noted in our response to Reviewer 3FCb, we conducted additional experiments replacing PAE’s evaluator with the one used in SAGE to isolate the effect of evaluator quality. The results show that even under the same evaluator, SAGE still outperforms PAE, confirming that the performance gain primarily stems from our system design:

| Model| Map  | Reddit | OSS   | Gitlab | CMS   | Avg  |
|---|----|----|---|---|---|---|
| Qwen2.5VL-7B PAE | 23.8 | 24.6   | 37.4  | 15.3   | 12.4  | 22.7 |
| Qwen2.5VL-7B PAE+SAGE's Eval | 24.1 | 24.4   | 37.6  | 14.9   | 12.9 | 22.9  |
| Qwen2.5VL-7B SAGE| 28.1 | 29.7   | 43.2  | 18.9   | 15.6  | 27.1 |
| Qwen2.5VL-32B PAE | 34.5 | 35.9   | 41.0  | 24.1   | 22.0  | 31.5 |
| Qwen2.5VL-32B PAE+SAGE's Eval| 35.8 | 36.2   | 43.1  | 26.4   | 23.2  | 33.0 |
| Qwen2.5VL-32B SAGE  | 42.3 | 43.7   | 46.1  | 31.4   | 35.2  | 39.7 |

These results reinforce that SAGE’s success is driven by its core innovations, including dynamic curriculum generation and hierarchical exploration. While a small amount of human-annotated data is used to fine-tune the evaluator, applying the same evaluator to PAE brings only marginal gains (average +0.8), compared to the much larger improvement from SAGE’s full pipeline (average +3.4). This demonstrates that **limited human supervision alone is insufficient**, and that SAGE’s design is critical to its performance.

Once again, we sincerely thank all reviewers, AC, and SAC for their time and support. We will incorporate all results and discussions into the final version.

---

### Decision · Program_Chairs · 2025-09-17

**Decision:**

Accept (poster)

**Comment:**

The submission proposes an algorithm for training web agents to solve tasks in a particular environment. First, it explores the website to build a map. Then, it generates tasks starting with lower complexity. Finally, MCTS search is used to improve the policy.

The reviewers concur on the value of the strong results. Most of the questions were clarified in the rebuttal.

For instance, reviewer 3FCb said:

> the method is very complex and the presentation is rather high-level, it would be very hard to reproduce the results from this paper. Additionally, the paper glosses over some crucial details, such as e.g. “a limited number of human-annotated examples” that were used to train the reward model.

Indeed, MCTS requires an evaluator. The supplementary material says that training the evaluator required manual annotation of 400 cases where Claude 3.5 Sonnet and humans disagreed.

One key issue was pointed out by reviewer ZYML, who, in their final justification, said:
> I still have concerns about the validity and quality of the generated tasks

The task generator is an LLM prompted to generate tasks of increasing difficulty. That probably worked because the benchmarks are well-known user experiences on the web.

Considering the empirical results, the quality of the ablation study, and despite the lack of analysis of the task-generated data, the anecdotal discussion of the evaluator's quality, and the possible missing details for reproducing the work, I consider that the submission should be accepted.

Please follow the recommendation of the Ethics Reviewer